# Sherlock: Self-Correcting Reasoning in Vision-Language Models

**Yi Ding, Ruqi Zhang**
Department of Computer Science, Purdue University, USA
`{ding432, ruqiz}@purdue.edu`

**Project Page:** https://dripnowhy.github.io/Sherlock/

## Abstract

Reasoning Vision-Language Models (VLMs) have shown promising performance on complex multimodal tasks. However, they still face significant challenges: they are highly sensitive to reasoning errors, require large volumes of annotated data or accurate verifiers, and struggle to generalize beyond specific domains. To address these limitations, we explore self-correction as a strategy to enhance reasoning VLMs. We first conduct an in-depth analysis of reasoning VLMs' self-correction abilities and identify key gaps. Based on our findings, we introduce *Sherlock*, a self-correction and self-improvement training framework. *Sherlock* introduces a trajectory-level self-correction objective, a preference data construction method based on visual perturbation, and a dynamic $\beta$ for preference tuning. Once the model acquires self-correction capabilities using only 20k randomly sampled annotated data, it continues to self-improve without external supervision. Built on the Llama3.2-Vision-11B model, *Sherlock* achieves remarkable results across eight benchmarks, reaching an average accuracy of 64.1 with direct generation and 65.4 after self-correction. It outperforms LLaVA-CoT (63.2), Mulberry (63.9), and LlamaV-o1 (63.4) while using less than 20% of the annotated data.

## 1 Introduction

Teaching Large Language Models (LLMs) to reason through long, step-by-step thinking processes during post-training has been shown to significantly improve their performance on complex tasks, such as mathematical and code benchmarks [13, 33, 11, 34]. Similarly, as Vision-Language Models (VLMs) demonstrate strong capabilities [18, 2, 59], recent studies have incorporated reasoning into VLMs through Supervised Fine-Tuning (SFT) [44, 35, 7, 48, 12, 31] or Reinforcement Learning (RL) [37, 38, 47, 54, 27, 26], boosting performance in solving challenging multimodal tasks.

Although reasoning VLMs such as [24, 31, 3, 32, 19] have demonstrated impressive performance on multimodal mathematical reasoning tasks, such as MathVista [21] and MathVerse [55], they still face significant challenges. First, they are highly sensitive to reasoning steps—once an error occurs in a multi-step reasoning process, it often propagates through subsequent steps, leading to incorrect final answers. Second, they are extremely data-demanding [12, 44], requiring large-scale, high-quality annotated data or easily verifiable answers to achieve consistent improvements. Third, they struggle to generalize to broader domains where such precise supervision is scarce, limiting their applicability beyond mathematical or VQA problem settings.

In this paper, we study *self-correction* as a promising strategy to address these challenges in reasoning VLMs. While self-correction has been explored in LLMs [14, 16], it remains underutilized in reasoning models or VLMs, despite being particularly well-suited. Self-correction allows the model to revise and improve its own prior outputs. This is particularly useful for challenging samples where the model may produce partially correct reasoning. In such cases, correcting specific mistakes through

---

39th Conference on Neural Information Processing Systems (NeurIPS 2025).

> **Case of Self-Correction**
>
> 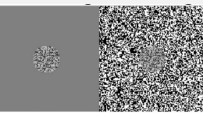 **Problem:** The image shows the famous Chubb illusion. Is the center area on the left smaller than the center area on the right?
>
> **Sherlock Direct Generation Response.** `<SUMMARY>` I will analyze the image to compare the size of the · · · . `</SUMMARY>`
> `<CAPTION>` The image is divided into two halves: the left side is a solid gray background with a small circular pattern in the center, while the right side features a black and white noise pattern with a central circle that is identical in size to the one on the left. `</CAPTION>`
> `<REASONING>` The central circle on both sides of the image is identical in size. However, the right side's central circle is embedded in a busy black and white pattern, which can create an illusion of size. · · · Despite this visual effect, the physical dimensions of the central circle do not change, and it remains smaller than the entire right half of the image. `</REASONING>`
> `<CONCLUSION>` Yes ✗ `</CONCLUSION>`
>
> **Sherlock Self-Correction Response.** `<SUMMARY>` I will analyze the image to determine the relative sizes of the · · · . `</SUMMARY>`
> `<CAPTION>` The image is divided into two halves. The left side shows a solid gray background with a small, circular, speckled pattern in the center. The right side features a dense, random black and white pattern covering the entire area, with a central circular region that is less dense. `</CAPTION>`
> `<REASONING>` To assess the size of the central areas, I will compare the two: the circular pattern on the left side is isolated and clearly defined against the gray background, suggesting a smaller presence. The central area on the right is defined by its lesser density within a sea of busy black and white patterns, suggesting a larger visual impact. **However, the physical dimensions of the two areas are the same.** `</REASONING>`
> `<CONCLUSION>` No ✓ `</CONCLUSION>`

Figure 1: Example of Sherlock's self-correction ability. The direct generation contains errors that cause the trajectory to deviate from the correct path and result in an incorrect answer. *Sherlock* successfully corrects the previous response and obtains the correct answer.

self-correction is simpler than generating a fully accurate answer from scratch [46]. Moreover, the responses before and after self-correction naturally form a preference pair [5, 52], where the inherent ordering acts as implicit supervision. By treating self-corrected outputs as preferred alternatives, the model not only reduces its dependency on extensive annotated data but also enhances its ability to generalize to broader domains.

To enable self-correction in reasoning VLMs, we introduce **Sherlock**: a self-correction and self-improvement training framework to enhance reasoning. Unlike existing self-correction approaches, *Sherlock* introduces a reasoning trajectory-level self-correction objective that focuses on correcting *only* the erroneous steps rather than the entire answer. Additionally, it leverages visual noise perturbations to construct preference datasets with controllable quality gaps. To further stabilize preference training, we introduce a dynamic $\beta$ that adapts to the quality gap between each sample pair. Once the model learns self-correction, it is able to self-improve without external supervision. Our main contributions are summarized as follows:

- We conduct an in-depth analysis of the self-correction capabilities of reasoning VLMs trained with either SFT or RL. We find that these models generally *cannot* self-correct. Self-correction within a reasoning trajectory was observed in fewer than 10% of samples, with only half leading to a correct final answer. When prompted with external critiques or self-correction prompts, the models fail to improve and may even see accuracy drop.

- To teach reasoning VLMs to self-correct, we propose a trajectory-level self-correction objective, a preference data construction pipeline, and a dynamic $\beta$ for preference tuning. This approach enables fine-grained correction at the reasoning trajectory level rather than overhauling the entire answer, providing a clearer learning signal.

- We present *Sherlock*, a training framework designed to enhance self-correction and reasoning in VLMs. It includes three stages: SFT, offline, and online preference learning. Through the entire training, *Sherlock* requires only **20k randomly sampled** annotated data. In the online stage, the model continues to improve both reasoning and self-correction abilities *without* any external supervision, leveraging self-constructed preference datasets.

- Across eight benchmarks, the *Sherlock* model achieves the best reasoning performance while using the least amount of annotated data (20k), reaching an average accuracy of 64.1 (direct reasoning) and 65.4 (after self-correction). In comparison, LLaVA-CoT (100k), Mulberry (260k), and LlamaV-o1 (175k) achieve 63.2, 63.9, and 63.4, respectively. To efficiently scale inference-time compute, we combine *Sherlock* with a verifier as a stopping criterion, reducing GPU usage by 40% while achieving even higher accuracy (54.0 → 55.9).

## 2 Related Works

**VLM Reasoning.** LLMs with reasoning abilities (i.e., thinking before answering) have been shown to significantly improve performance across diverse tasks [40, 13, 11]. Building on this insight, researchers are now equipping VLMs with deeper reasoning capabilities to enhance their performance on visual tasks. Reasoning abilities are primarily enabled by supervised fine-tuning (SFT) or reinforcement learning (RL) methods. SFT uses external supervision signals combined with Chain-of-Thought (CoT) [40] to teach models think step-by-step [56, 44, 12, 7, 6, 48, 35]. However, generating multimodal CoT annotations often involves multi-turn prompting or Monte Carlo Tree Search (MCTS) on powerful non-reasoning VLMs, resulting in substantial computational costs. RL methods elicit reasoning behaviors through reward maximization. Inspired by DeepSeek-R1 [11], recent studies [54, 3, 24, 27, 26, 31, 38, 37, 31, 19, 58] use carefully designed, rule-based rewards to induce multimodal "aha moments". Despite notable improvements, particularly in mathematical reasoning, these approaches remain highly dependent on carefully curated multimodal datasets and accurate verifiers. Training samples are often filtered based on complexity and reasoning difficulty, making these methods resource-intensive and challenging to generalize to diverse domains [24, 38]. In contrast, our method improves VLM reasoning by requiring only 20k randomly sampled CoT-labeled data to start, and then self-improves without relying on ground truth labels.

**Self-correction.** Self-correction refers to the model's ability to revise its previous responses to generate higher-quality answers. It has emerged as an intuitive and promising approach for enhancing model capabilities through inference-time scaling [22, 23, 57]. Studies [36, 28, 52] show that directly prompting LLMs to perform self-correction using generic instructions often leads to degraded performance. To address this, SFT and RL approaches [39, 49, 5, 16, 52] are developed to equip models with critique and self-correction capabilities. Step-wise correction methods [48, 37] use SFT to teach models to reflect and revise within a single reasoning process, but often fail to effectively trigger intrinsic correction behavior. The most relevant to our work is [43], which integrates response-wise correction into LLMs for mathematical reasoning tasks using correction prompts and trains with DPO and PPO. However, this approach is limited to domains with rule-verifiable answers and depends on such rules to construct self-correction datasets. In multimodal settings, some approaches train an additional critic model to generate sample-specific critiques [42, 53, 31]. However, deploying these extra models and collecting large-scale critique data are resource-intensive, and their effectiveness in judging long CoT responses remains uncertain. Unlike prior work, *Sherlock* is specifically designed for multimodal reasoning models. It introduces a trajectory-level self-correction objective and a corresponding data construction pipeline that guides the model to revise only the erroneous suffix of the reasoning trajectory, removing the need for verifiers or large-scale critique data.

## 3 Can Reasoning VLMs Self-Correct?

Reasoning VLMs explicitly generate step-by-step thoughts along with the final answer during inference [44, 35, 48, 12, 38, 47, 54, 37]. This process can be denoted as $(y_1, \cdots, y_n; a) \sim \pi(\cdot | x_{I\&T})$, where $y_i$ represents the $i$-th reasoning step, $a$ is the final answer, $\pi$ is the reasoning VLM, and $x_{I\&T}$ denotes the input image and text. Given language models' auto-regressive decoding mechanism, an error in any reasoning step $y_i$ may propagate and result in incorrect reasoning in the subsequent steps. Therefore, equipping reasoning models with self-correction capabilities holds great potential for improving overall reasoning quality. For reasoning models, self-correction behavior can be implemented in two ways: (i) **Step-wise correction**: The model reflects on its previous incorrect $i$-th step *within* a single thinking process and revises it to arrive at the final answer:

$$(r, y_{i+1}, \cdots, y_n; a) \sim \pi(\cdot | x_{I\&T}; y_1, \cdots, y_i^*), r \in \{\text{"wait", "however", "check", } \cdots\}. \quad (1)$$

Here, $y^*$ is the erroneous reasoning step, and $r$ is a reflection token indicating the model's intention to correct its previous reasoning. (ii) **Response-wise correction**: The model is prompted to revise and improve its previously generated response:

$$(y_1^2, \cdots, y_n^2; a^2) \sim \pi(\cdot | x_{I\&T}; y_1^1, \cdots, y_n^1, a^1; t), \quad (2)$$

where $\{y_i^j, a^j\}$ denotes the $j$-th attempt and $t$ is an additional instruction guiding the model to perform correction. In this section, we analyze both types of self-correction behaviors of reasoning VLMs, including the SFT-based LLaVA-CoT [44] and the RL-based VL-Rethinker [37], on two widely-used benchmarks, MMStar [4] and MathVista [21].

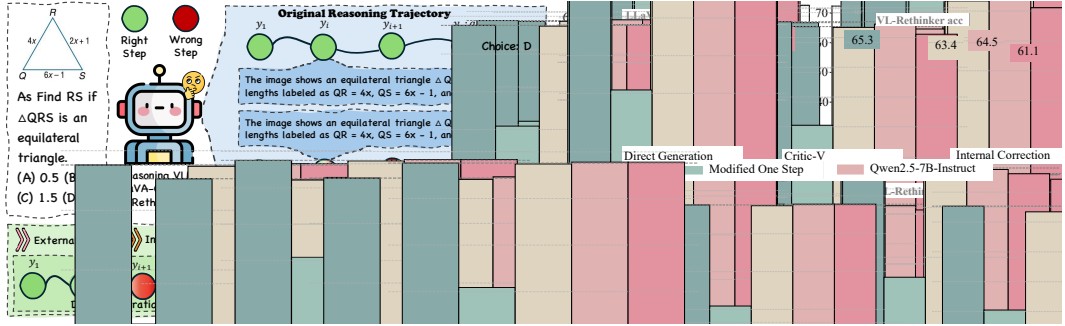

Figure 2: **Left**: Overview of experimental settings for self-correction analysis. Blue block illustrates the Modified One Step process using Qwen2.5-7B-Instruct [45], while Green block represents two correction strategies applied to direct generations: external critique-based correction and self-correction prompt. **Right**: Reasoning performance of LLaVA-CoT [44] and VL-Rethinker [37] under different settings, evaluated on MMStar [4] and MathVista [21].

## 3.1 Step-wise Self-correction

**Setup.** To test whether the model can perform step-wise self-correction, we conduct a controlled experiment using Qwen2.5-7B-Instruct [45]. In this experiment, we prompt the Qwen model to autonomously select one influential step $y_i$ in the first half of the original reasoning trajectory and apply **Modify One Step**, resulting in an erroneous step denoted as $y_i^*$. Then, VLMs generate the subsequent trajectory $(y_{i+1}^*, \cdots, y_n^*; a^*) \sim$

Table 1: Step-wise self-correction results of reasoning VLMs under the **Modify One Step** setting. The numbers following the benchmark names (1500 and 1000) indicate the total number of samples in each benchmark.

| Model | MMStar (1500) w/ Aha Moment | | MathVista (1000) w/ Aha Moment | |
|---|---|---|---|---|
| | Num | Acc | Num | Acc |
| LLaVA-CoT [44] | 121 (8.1%) | 47.1 | 104 (10.4%) | 27.9 |
| VL-Rethinker-7B [37] | 136 (9.1%) | 52.2 | 94 (9.4%) | 63.8 |

$\pi(\cdot|x_{I\&T}; y_1, \cdots, y_{i-1}, y_i^*)$ conditioned on the previous reasoning steps with incorrect $y_i^*$. This process is illustrated in the blue block of Fig. 2. To examine whether models exhibit step-wise self-correction behavior, which is often accompanied by the emergence of an "*aha moment*", and whether such behavior contributes to successful correction and the correct final answer, we analyze whether the suffix $(y_{i+1}^*, \cdots, y_n^*)$ contains indicative expressions such as "wait", "however", or "check", which we denote as **w/ Aha Moment**. We report both the overall accuracy under the **Modify One Step** setting and the accuracy within the subset of responses that exhibit an **Aha Moment**.

**Findings.** We report the performance of reasoning VLMs under **Modify One Step** in the right part of Fig. 2. By comparing direct generation with the Modify One Step setting, we observe that an error in a single step often leads to an incorrect final answer, causing the overall accuracy to drop to the level of a random guess (∼25%). Additionally, Table 1 shows the number of responses exhibiting "*aha moments*" (i.e., signs of self-reflection) after an error step. The results indicate that even the RL-based model VL-Rethinker exhibits "*aha moments*" in fewer than 10% of cases. Moreover, the presence of self-reflective behavior does not necessarily lead to a correct final answer. More details about the settings are provided in Appendix C.3.

> **Takeaway 1: Step-wise Self-Correction**
>
> Reasoning VLMs struggle with step-wise self-correction. Even when reflection signals are present, they often fail to correct their reasoning to reach the correct answer.

## 3.2 Response-wise Self-correction

**Setup.** To analyze the models' response-wise self-correction behavior, we evaluate two strategies: (i) applying an **self-correction prompt** to guide the model in refining its responses; (ii) leveraging **external critiques** generated by critic models, such as Critic-V [53] and Qwen2.5-VL-7B-Instruct [2], to prompt VLMs to improve their responses based on critique feedback. The evaluation procedure

consists of: (1) generating the initial **Direct Generation** response; and (2) applying two correction strategies to obtain revised responses, reported as **Internal Correction** and **External Critique**, with the latter involving two different critique models, as illustrated in the green block of Fig. 2. Complete prompts and more details are shown in Appendix C.3.

**Findings.** The results in the right part of Fig. 2 show that neither the correction prompt nor the external critiques effectively improve the reasoning trajectory. In some cases, model accuracy even declines after correction. Moreover, we observe that, regardless of which critic model is used to provide feedback, the accuracy of the corrected responses tends to converge to a level similar to direct generation, showing little variation based on the quality of the critique.

> **Takeaway 2: Response-wise Self-Correction**
>
> Reasoning VLMs struggle with response-wise self-correction, regardless of whether using self-correction prompts or external critiques.

**Step-wise vs. Response-wise Self-Correction** Step-wise self-correction depends on the internal reflection during intermediate reasoning, which is hard to control or reliably trigger. In contrast, response-wise self-correction is externally guided (e.g., by self-correction prompts), making it more controllable and learnable. Hence, this work focuses on teaching VLMs response-wise self-correction.

## 4 Teaching VLMs to Self-Improve Their Self-Correction Abilities

Takeaway 1 and 2 reveal a critical gap in current reasoning VLMs: models trained with either SFT or RL lack the ability to perform effective step-wise and response-wise self-correction. Once an error occurs, the models struggle to revise their reasoning, often failing to recover from mistakes. To address this limitation, we introduce *Sherlock* to teach the model self-correction, thereby enhancing its reasoning capabilities. The framework consists of three stages, which we will detail below.

### 4.1 Stage I: SFT Cold-start for Reasoning and Self-correction

Recent studies [44, 48, 12, 35] have shown that supervised fine-tuning VLMs on long CoT annotated data can significantly enhance their general reasoning capabilities. To teach VLMs to self-correct, we draw inspiration from [52, 48] and aim to design an objective that simultaneously improves reasoning and self-correction. To achieve this, we introduce a pairwise training objective:

$$\mathcal{L}_{\text{Sherlock-SFT}}(\pi) = -\mathbb{E}_{(x_{I\&T}, Y^w, Y^l) \sim \mathcal{D}_{\text{Sherlock}}} [\underbrace{\log \pi(Y^w | x_{I\&T})}_{\text{Direct Generation}} + \underbrace{\log \pi(Y^w | x_{I\&T}, Y^l, t)}_{\text{Self-Correction}}]. \quad (3)$$

We begin by randomly sampling **10k** examples from the LLaVA-CoT dataset [44], denoted as $\mathcal{D}_A$. We then train a base VLM on $\mathcal{D}_A$ using vanilla SFT, resulting in *R0 VLM*, which is capable of generating CoT template responses. Next, we randomly sample an additional **10k** examples, denoted as $\mathcal{D}_B$. Since $\mathcal{D}_B$ is annotated with high-quality reasoning trajectories, we assume its responses are of higher quality than those sampled from the model fine-tuned on limited data, i.e., $Y \sim \pi_{\text{R0 VLM}}(\cdot | x_{I\&T})$. We then construct a Sherlock-SFT dataset $\mathcal{D}_{\text{Sherlock}} = (x_{I\&T}^i, Y_i^w, Y_i^l)_{i=1}^{10k}$, where $Y^w$ is from $\mathcal{D}_B$ and $Y^l$ is generated by $\pi_{\text{R0 VLM}}$. Finally, we apply the loss defined in Eq. 3 to cold-start the base VLM, jointly training the *Sherlock SFT* model to acquire both reasoning and self-correction capabilities.

### 4.2 Stage II: Offline Preference Training with Trajectory-level Self-correction

An incorrect reasoning process may contain partially correct steps, but the loss in Eq. 3 enforces correction over the entire response, potentially introducing noise by forcing the model to revise already correct steps. To avoid such undesirable behavior and inspired by Takeaway 1, which shows that early mistakes can derail the entire reasoning trajectory, we adopt a fine-grained trajectory-level offline training strategy. Instead of rewriting the full response, the model is guided to revise only the erroneous suffix, allowing it to make targeted adjustments while preserving correct reasoning steps. Our goal is to enable reasoning VLMs to self-correct faulty steps $Y_{\geq i}^1 = (y_i^1, \cdots, y_n^1; a^1)$ in the initial $Y^1 = (y_1^1, \cdots, y_n^1; a^1)$ and produce a higher-quality trajectory $Y^2 = (y_1^2, \cdots, y_n^2; a^2)$, conditioned on input $x_{I\&T}$, a correction instruction $t$, and the original response $Y^1$. Since the prefix

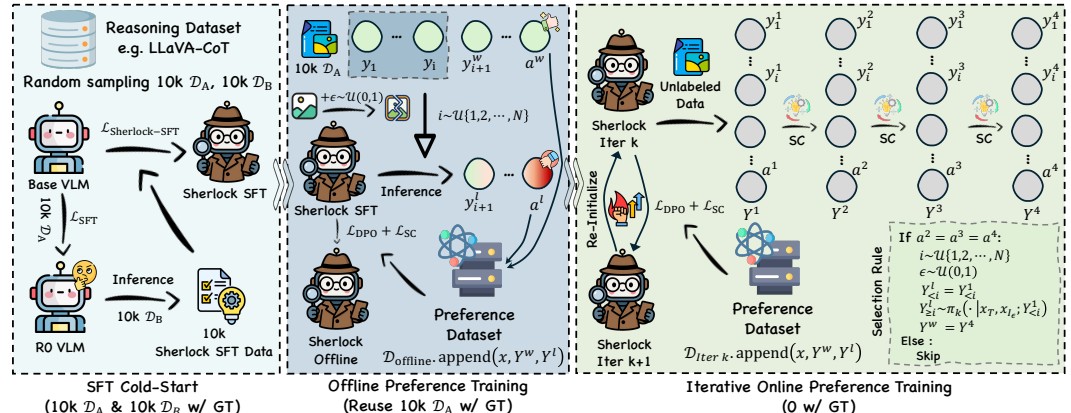

Figure 3: Training pipeline of *Sherlock*, including: (Left) SFT cold-start stage, (Middle) offline preference training, and (Right) online iterative self-improvement. In the SFT and offline stages, we randomly sample 10k $\mathcal{D}_A$ and 10k $\mathcal{D}_B$ with ground truth from the 100k LLaVA-CoT [44] dataset as supervision. During the online stage, each iteration samples only 5k unlabeled inputs, from which a self-constructed and self-labeled dataset is built using the selection rule illustrated in the (Right) part.

$Y^1_{<i}$ is assumed to be correct, we do not enforce revisions on these steps. The objective instead focuses on generating a higher-quality suffix $Y^2_{\geq i}$:

$$\max_{\pi} \mathbb{E}_{Y^2_{\geq i} \sim \pi(\cdot|[x_{I\&T}, Y^1, t; Y^2_{<i}])} \left[ p(Y^2_{\geq i} \succ Y^1_{\geq i}|x_{I\&T}; Y^2_{<i}) - \beta D_{\mathrm{KL}}(\pi\|\pi_{\mathrm{ref}}|[x_{I\&T}, Y^1, t; Y^2_{<i}]) \right]$$

$$+ \mathbb{E}_{Y^2_{\geq i} \sim \pi(\cdot|[x_{I\&T}, Y^1, t; Y^1_{<i}])} \left[ p(Y^2_{\geq i} \succ Y^1_{\geq i}|x_{I\&T}; Y^1_{<i}) - \beta D_{\mathrm{KL}}(\pi\|\pi_{\mathrm{ref}}|[x_{I\&T}, Y^1, t; Y^1_{<i}]) \right]. \quad (4)$$

Here, $i$ is randomly sampled from 1 to $n$ for each example. The first expectation encourages the model to prefer the higher-quality suffix $Y^2_{\geq i}$ over $Y^1_{\geq i}$, given the prefix $Y^2_{<i}$, while remaining close to the reference policy $\pi_{\mathrm{ref}}$. The second expectation does the same but conditioned on $Y^1_{<i}$. This objective has two key benefits: it leverages both prefixes $Y^1_{<i}$ and $Y^2_{<i}$, which contain no explicit preference signals, and it explicitly models the preference between $Y^2_{\geq i}$ and $Y^1_{\geq i}$, promoting correct self-correction while discouraging incorrect updates. Following prior work [25, 5, 1, 52], we denote $p(Y^{\geq i}_2 \succ Y^{\geq i}_1|x_{I\&T}; Y^{<i})$ as the human preference probability over suffix reasoning trajectories given the instruction and prefix, expressed as:

$$p(Y^2_{\geq i} \succ Y^1_{\geq i}|x_{I\&T}; Y^j_{<i}) = \mathbb{E}_{\mathrm{h}}[\mathbb{I}\{\mathrm{h \ prefers \ } Y^2_{\geq i} \mathrm{\ to \ } Y^1_{\geq i} \mathrm{\ given \ } x_{I\&T} \mathrm{\ and \ } Y^j_{<i}\}], j \in \{1, 2\}. \quad (5)$$

Building upon the formulation in [5, 52], we extend it to the trajectory-level and solve the objective in Eq. 4, resulting in the following self-correction loss:

$$v(x, Y^1, t; Y^1_{<i}, Y^2_{<i}; \pi_\theta) = \underbrace{\beta \log \frac{\pi_\theta(Y^2_{\geq i}|[x, Y^1, t; Y^2_{<i}])}{\pi_{\mathrm{ref}}(Y^2_{\geq i}|[x, Y^1, t; Y^2_{<i}])}}_{\text{Encourage positive self-correction}} \underbrace{-\beta \log \frac{\pi_\theta(Y^1_{\geq i}|[x, Y^1, t; Y^1_{<i}])}{\pi_{\mathrm{ref}}(Y^1_{\geq i}|[x, Y^1, t; Y^1_{<i}])}}_{\text{Discourage negative self-correction}}$$

$$u(x, Y^1, t; Y^1_{<i}, Y^2_{<i}; \pi_\theta) = \underbrace{\beta \log \frac{\pi_\theta(Y^2_{\geq i}|[x, Y^1, t; Y^1_{<i}])}{\pi_{\mathrm{ref}}(Y^2_{\geq i}|[x, Y^1, t; Y^1_{<i}])}}_{\text{Encourage positive self-correction}} \underbrace{-\beta \log \frac{\pi_\theta(Y^1_{\geq i}|[x, Y^1, t; Y^2_{<i}])}{\pi_{\mathrm{ref}}(Y^1_{\geq i}|[x, Y^1, t; Y^2_{<i}])}}_{\text{Discourage negative self-correction}}$$

$$\mathcal{L}_{\mathrm{SC}}(\pi_\theta; \pi_{\mathrm{ref}}) = \mathbb{E}_{(x, Y^w, Y^l) \sim \mathcal{D}} \left[ 1 - v(x_{I\&T}, Y^l, t; Y^l_{<i}, Y^w_{<i}; \pi_\theta) - u(x_{I\&T}, Y^l, t; Y^l_{<i}, Y^w_{<i}; \pi_\theta) \right]^2$$

$$+ \mathbb{E}_{(x, Y^w, Y^l) \sim \mathcal{D}} \left[ 1 + v(x_{I\&T}, Y^w, t; Y^w_{<i}, Y^l_{<i}; \pi_\theta) + u(x_{I\&T}, Y^w, t; Y^w_{<i}, Y^l_{<i}; \pi_\theta) \right]^2. \quad (6)$$

The detailed derivation and explanation are provided in Appendix A. To further enhance the model's direct reasoning ability using the preference dataset, we additionally incorporate the DPO loss [29], which is jointly optimized with the self-correction loss:

$$\mathcal{L}_{\mathrm{Sherlock}}(\pi_\theta; \pi_{\mathrm{ref}}) = \mathcal{L}_{\mathrm{SC}}(\pi_\theta; \pi_{\mathrm{ref}}) + \alpha \mathcal{L}_{\mathrm{DPO}}(\pi_\theta; \pi_{\mathrm{ref}}) \quad (7)$$

**Data Construction.** We construct negative responses with controlled quality gaps by randomly truncating reasoning steps and adding visual noise. From the 10k annotated examples in $\mathcal{D}_A$, we

sample a truncation point $i \sim \mathcal{U}\{1, \cdots, N\}$ and generate a suffix $Y^l \geq i$ by applying visual perturbation $\epsilon$ and decoding from $Y^l_{<i}$. This produces low-quality reasoning trajectories for preference training. The model is then trained on this dataset using Eq. 7 to obtain the *Sherlock Offline* model. The complete data construction process is described in Appendix B.2.

**Dynamic $\beta$ Stabilize Preference Training.** Prior work shows that the choice of $\beta$ strongly affects preference optimization [5, 52, 41]. In our framework, we build a controllable preference dataset by randomly selecting a truncation step $i$ and applying visual perturbation $\epsilon$, enabling a sample-specific dynamic $\beta$ to adaptively scale learning signals:

$$\beta(i, n, \epsilon) = \frac{1}{4(0.5 + \left(\frac{i}{n}\right)^{0.5 + \epsilon/2})}. \tag{8}$$

Larger $\beta$ values are assigned to samples with earlier truncation or stronger perturbation, where quality gaps are wider, yielding more conservative updates. Smaller gaps result in smaller $\beta$, promoting stronger learning from subtle preferences. Further details on $\beta$ design are provided in Appendix B.2.

> **Takeaway 3: Key of Sherlock Objective**
>
> The key insight of *Sherlock* is to **fully leverage** the constructed trajectory-level preference data and **explicitly** encourage models to revise only the incorrect suffix reasoning trajectory.

### 4.3 Stage III: Iterative Online Preference Training with Self-generated Data

The *Sherlock* model has acquired both self-correction and reasoning capabilities using only a small amount of data in the SFT and offline stages. Inspired by recent LLMs self-improvement works [52, 43, 5], we observe that the original and corrected responses naturally form preference pairs. Motivated by this, we develop a self-improvement framework to further enhance the model's capabilities.

The online iterative training differs from the offline stage (Sec. 4.2) only in the absence of ground-truth responses $Y^w$. As previously discussed, reasoning quality improves through sequential self-correction. In each iteration, we randomly sample **5k unlabeled** questions. For each initial generation $Y^1$, three rounds of self-correction produce $Y^2$, $Y^3$, and $Y^4$. We then apply self-consistency filtering: if the final answers are semantically identical ($a^2 = a^3 = a^4$), $Y^4$ is selected as the preferred response. To further reduce noise, the rejected response is derived from $Y^1$ by keeping the prefix $Y^l_{<i} = Y^1_{<i}$ and generating a perturbed $Y^l_{\geq i}$ via visual noise injection, following the offline stage. We perform two iterations of self-improvement training using Eq. 7, resulting in the *Sherlock Iter1* and *Iter2* models.

> **Takeaway 4: Self-Improvement Training Enabled by Self-Correction**
>
> When the model can self-correct, it naturally generates a pair of responses—the original and the corrected versions—with a clear quality preference, enabling iterative self-improvement.

## 5 Experiments

### 5.1 Setup

**Training Details.** Following prior SFT-based methods [44, 48, 35], Sherlock builds on Llama3.2V-11B-Instruct [9], focusing on integrating self-correction and reasoning capabilities. Additional training details, including data construction and hyperparameters, are provided in Appendix B.

**Baselines.** We evaluate *Sherlock* from two aspects: reasoning and self-correction. For reasoning, we compare against LLaVA-CoT [44], Mulberry [48], and LlamaV-o1 [35], all built on Llama3.2-Vision-11B-Instruct [9]. For self-correction, we include inference-time scaling baselines such as Majority Vote and critique-based methods using LLaVA-Critic [42], Critic-V [53], and Qwen2.5-VL-7B-Instruct [2]. Details of these baselines are provided in Appendix C.2.

**Evaluation Details.** We evaluate on eight challenging multimodal benchmarks, including VQA (MMBench-V1.1 [20], MMVet [50], MME [17], MMStar [4]), math and science (MathVista [21], AI2D [15], MMMU [51]), and hallucination (HallusionBench [10]). *Sherlock* is tested under two settings: direct generation and after three rounds of self-correction. All evaluations are conducted using the VLMEvalKit [8] for consistency. Additional details are provided in Appendix C.1 and C.4.

Table 2: Performance comparison across 8 benchmarks. #Data w/ GT indicates the number of ground-truth annotated samples used during training. MMB refers to MMBench-V1.1, Hallus denotes HallusionBench, and MathV corresponds to MathVista. **Bold** and underline indicate the best and second-best results, respectively. The red number represent the accuracy change after self-correction.

| Models | #Data w/ GT | MMB | MMVet | Hallus | MMMU | MMStar | AI2D | MathV | MME | Avg. |
|---|---|---|---|---|---|---|---|---|---|---|
| Llama3.2V-11B-Ins [9] | - | 65.8 | 57.6 | 42.7 | 47.8 | 53.0 | 88.2 | 49.7 | 1822 | 58.7 |
| ***Reasoning Models*** | | | | | | | | | | |
| LLaVA-CoT [44] | 100k | 75.0 | 61.7 | 47.7 | 49.1 | 57.6 | 82.9 | 53.7 | 2177 | 63.2 |
| + Self-Correction | | 74.4 | 62.3 | 46.4 | 49.2 | 57.8 | 82.9 | 53.0 | 2183 | 63.0$^{0.2\downarrow}$ |
| Mulberry [48] | 260k | 75.2 | 58.3 | 47.8 | 46.7 | 57.8 | 86.2 | 61.9 | 2170 | 63.9 |
| + Self-Correction | | 74.2 | 59.0 | 46.6 | 46.9 | 57.4 | 86.3 | **62.3** | 2177 | 63.8$^{0.1\downarrow}$ |
| LlamaV-o1 [35] | 175k | 75.6 | 61.9 | 45.6 | **52.3** | 56.5 | 86.4 | 53.3 | 2125 | 63.4 |
| + Self-Correction | | 18.4 | 50.9 | 39.4 | 43.9 | 47.1 | 76.9 | 44.0 | 1823 | 48.2$^{15.2\downarrow}$ |
| ***Ours Sherlock Models*** | | | | | | | | | | |
| *Sherlock SFT* | 10k | 72.2 | 61.4 | 45.5 | 47.1 | 54.9 | 86.6 | 52.0 | 2170 | 62.2 |
| + Self-Correction | | 73.8 | 62.8 | 47.5 | 46.2 | 55.9 | 87.9 | 52.2 | 2172 | 63.0$^{0.8\uparrow}$ |
| *Sherlock Offline* | 10k | 73.2 | 61.4 | 48.1 | 47.6 | 57.5 | 88.4 | 52.2 | 2162 | 63.2 |
| + Self-Correction | | 74.7 | **63.8** | 48.9 | 49.0 | 57.7 | 89.5 | 53.9 | 2171 | 64.4$^{1.2\uparrow}$ |
| *Sherlock Iter1* | 0 | 74.9 | 62.3 | 49.7 | 48.2 | 57.0 | 88.9 | 52.2 | 2177 | 63.9 |
| + Self-Correction | | 76.6 | 62.7 | 50.6 | 49.2 | 58.8 | 90.0 | 54.4 | 2195 | 65.1$^{1.2\uparrow}$ |
| *Sherlock Iter2* | 0 | 74.6 | 62.4 | 48.7 | 49.7 | 57.7 | 89.6 | 52.0 | 2197 | 64.1 |
| + Self-Correction | | **77.2** | 62.6 | **51.2** | 50.1 | **59.0** | **90.6** | 54.0 | **2204** | 65.4$^{1.3\uparrow}$ |

## 5.2 Main Results

**Advancing Reasoning Ability with Minimal Annotation.** The results in Table 2 show that *Sherlock Iter2*, trained on only 20k annotated examples randomly sampled from LLaVA-CoT, achieves the best overall reasoning performance. It outperforms LLaVA-CoT [44] (100k annotations), Mulberry [48] (260k), and LlamaV-o1 [35] (170k) across 8 benchmarks. Notably, even before online self-improvement, *Sherlock Offline* performs comparably to LLaVA-CoT, highlighting that incorporating self-correction significantly enhances direct generation quality. During online training, both reasoning accuracy (63.2→63.9→64.1) and self-correction gains (1.2↑→1.3↑) steadily improve, demonstrating the effectiveness of self-generated preference data and self-improvement training.

**Existing Models Fail to Self-correct Their Reasoning.** Table 2 shows the performance of reasoning VLMs after three rounds of self-correction. Consistent with Takeaway 2, existing reasoning models fail to improve through internal prompt-based self-correction and even show performance drops across eight benchmarks. Notably, LlamaV-o1 degrades sharply because its step-by-step reasoning is generated via multi-turn prompting; adding a correction prompt after the final turn breaks its original training format, leading to poorer responses.

**Sherlock Further Enhances Performance via Self-Correction.** Table 2 also presents the results of three-round self-correction on *Sherlock*. During SFT, we use only a response-wise self-correction objective without explicitly revising incorrect trajectories ($Y_{\geq i}^l$). After both offline and online training, the model shows clear gains over *Sherlock SFT*, validating the effectiveness of our trajectory-level objective. Table 3 compares different inference-time scaling strategies. Except for Majority Vote, all methods rely on external critics for response critiques. Unlike in Sec. 3.2, we find critique-based correction highly dependent on critic quality: the strong Qwen2.5-VL-7B-Instruct yields the best results, while weaker critics cause degradation, likely due to difficulty processing long CoT outputs. This sensitivity highlights *Sherlock*'s learned responsiveness to contextual feedback. Notably, self-correction outperform Majority Vote while requiring only half the inference time.

## 5.3 Ablation Studies on Training Objective

We ablate our *Sherlock* pipeline, including the combination of DPO loss and self-correction loss in Eq. 7, the selection strategy for the prefix step $i$ in Eq. 6, and the design of the dynamic $\beta$ in Eq. 8.

Table 3: Performance of inference-time methods. LLaVA-Critic, Critic-V, and Qwen2.5-VL-7B provide critiques for correction, while Majority Vote @8 selects an answer from 8 sampled generations.

| Methods | MMB | MMVet | Hallus | MMMU | MMStar | AI2D | MathV | MME | Avg. |
|---|---|---|---|---|---|---|---|---|---|
| *Sherlock Iter2* | 74.6 | 62.4 | 48.7 | 49.7 | 57.7 | 89.6 | 52.0 | 2197 | 64.1 |
|   + LLaVA-Critic [42] | 75.5 | 58.9 | 45.9 | 47.0 | 58.7 | 89.1 | 52.6 | 2122 | 62.9[1.2↓] |
|   + Critic-V [53] | 73.9 | 61.8 | 47.0 | 47.7 | 58.1 | 88.9 | 50.2 | 2192 | 63.2[0.9↓] |
|   + Qwen2.5-VL-7B [2] | 76.5 | **64.4** | 48.6 | 47.9 | **59.3** | 89.1 | **55.5** | 2189 | 64.9[0.8↑] |
|   + Majority Vote @8 | **78.5** | 62.2 | 49.3 | 49.7 | 58.0 | **91.1** | 54.0 | 2195 | 65.1[1.0↑] |
|   + Self-Correction | 77.2 | 62.6 | **51.2** | 50.1 | 59.0 | 90.6 | 54.0 | 2204 | **65.4**[1.3↑] |

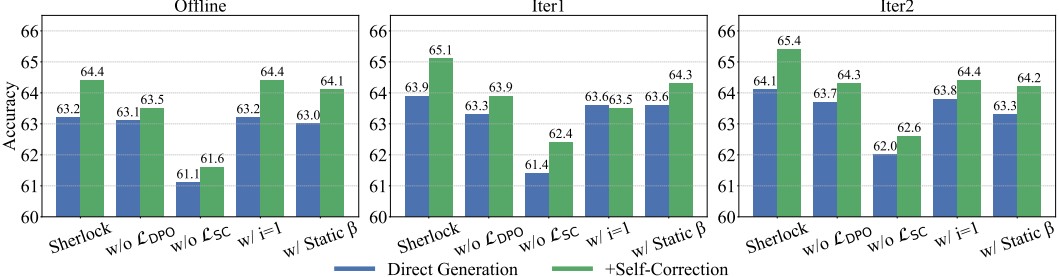

Figure 4: Average accuracy across 8 benchmarks for ablation settings. w/ i=1 indicates that the objective in Eq. 6 performs self-correction on the entire response instead of trajectory-level.

**Finding 1: Self-correction and Reasoning are Not Orthogonal Capabilities.** As shown in Fig. 4, we compare the full *Sherlock* model with two ablated variants trained using only the DPO loss or only the self-correction loss. We observe that using only the DPO loss brings minimal improvement over *Sherlock-SFT*. In contrast, the model trained solely with the self-correction objective continues to improve across iterations. Although its overall performance is slightly lower than the full *Sherlock* model, it achieves direct reasoning accuracy comparable to other baselines, despite receiving no explicit supervision for direct generation. This finding aligns with prior work showing that critique-based supervision can benefit direct reasoning [39]. Interestingly, even the DPO-only model shows gains in self-correction over iterations, despite having access to self-correction signals only during the initial SFT stage. These results collectively suggest that self-correction and reasoning are not independent abilities, but mutually reinforcing: learning one promotes the other.

**Finding 2: Trajectory-level Objective Yields Superior Self-correction in Reasoning VLMs.** In Fig. 4, we compare different strategies for selecting step $i$ in Eq. 6. Setting $i = 1$ reduces the loss to full-response correction, forcing revision over the entire response. In contrast, *Sherlock* adopts trajectory-level correction, updating only the low-quality suffix $Y_{\geq i}^l$ while leaving prefixes $Y_{<i}^l$ and $Y_{<i}^w$ untouched, as they contain no clear preference signals. The offline model remains stable since both preferred and rejected responses share identical prefixes, contributing no gradients. However, in online iterations, full-response correction degrades self-correction performance, as strong preference signals mainly lie in $Y_{\geq i}^l$, and updating non-informative prefixes hinders learning.

**Finding 3: Dynamic $\beta$ Leads to Stable Training.** We present results in Fig. 4 comparing whether the dynamic $\beta$ is applied to Eq. 6. Experiments show that incorporating dynamic $\beta$ consistently improves both direct generation and self-correction performance across iterations. These findings validate that our sample-specific dynamic $\beta$ contributes to more stable training and stronger models.

## 5.4 Why Does Sherlock Need Less Annotated Data?

Sherlock requires only 20k randomly sampled annotated examples because it fully leverages each example through its training objectives. To highlight Sherlock's data efficiency, we compare it against two alternative training strategies under the same 20k data budget: (i) training with standard SFT loss as adopted in LLaVA-CoT, without any self-correction objective, and (ii) training only using Sherlock-SFT loss defined in Eq. 3, without applying the offline preference training objective.

Table 4: Performance comparison of different methods using the same 20k annotated data.

| Methods | MMB | MMVet | Hallus | MMMU | MMStar | AI2D | MathV | MME | Avg. |
|---|---|---|---|---|---|---|---|---|---|
| LLaVA-CoT 20k | 72.0 | 58.9 | 47.9 | 47.9 | 55.3 | 88.3 | 48.7 | 2158 | 62.0 |
| + Self-Correction | 71.6 | 58.6 | 46.8 | 48.7 | 56.0 | 87.7 | 49.8 | 2109 | $61.8^{0.2\downarrow}$ |
| *Sherlock SFT 20k* | 72.1 | 61.2 | 47.6 | 45.6 | 57.0 | 88.0 | 52.5 | **2186** | 62.8 |
| + Self-Correction | 74.1 | 62.2 | 47.8 | 46.0 | **58.1** | 89.0 | 53.2 | 2159 | $63.4^{0.6\uparrow}$ |
| *Sherlock Offline* | 73.2 | 61.4 | 48.1 | 47.6 | 57.5 | 88.4 | 52.2 | 2162 | 63.2 |
| + Self-Correction | **74.7** | **63.8** | **48.9** | **49.0** | 57.7 | **89.5** | **53.9** | 2171 | $\mathbf{64.4}^{1.2\uparrow}$ |

**Integrating Self-correction into SFT: A Free Lunch for Enhancing Reasoning.** Under the same 20k data setting, the only difference between Sherlock SFT and LLaVA-CoT is the self-correction term in Eq. 3. As shown in Table 4, Sherlock achieves 0.8 higher average accuracy in direct reasoning, consistent with **Finding 1** (Sec. 5.3) that learning self-correction enhances reasoning ability. After applying self-correction, Sherlock further improves, widening the gap over LLaVA-CoT.

**Trajectory-level Objective Maximizes the Use of Preference Data.** Table 4 shows that *Sherlock-Offline* consistently outperforms *Sherlock-SFT*, both in terms of direct reasoning performance (63.2 & 62.8) and improvements (1.2↑ & 0.6↑) brought by self-correction. While the *Sherlock-SFT* loss (Eq. 3) focuses solely on guiding the model to revise low-quality responses into high-quality ones, the *Sherlock-Offline* loss (Eq. 6) incorporates three additional objectives: (1) discouraging the model from generating the same low-quality response again, (2) preventing quality degradation when the initial response is already high-quality, and (3) encouraging the model to reproduce the same output when the initial reasoning is correct. These complementary signals help the model fully utilize the preference data to learn more fine-grained and robust self-correction behavior.

## 5.5 Efficiently Scaling Up Sequential Self-Correction with Verifiers

Recalling the results in Sec. 5.2, where we validated that the *Sherlock* model achieves stable inference-time scaling through self-correction. In this section, inspired by recent work MM-Verify [31], we conduct a deeper exploration aimed at simultaneously improving the efficiency and capability of self-correction-based inference-time scaling.

Table 5: Inference-time scaling results with MM-Verify on MathVista.

| Methods | Acc | GPU Hours |
|---|---|---|
| *Sherlock Iter2* | 52.0 | 3.3 |
| + Self-Correction w/o Verify | 54.0 | 13.2 |
| + Parallel Vote w/ Verify | 55.1 | 40.2 |
| + Self-Correction w/ Verify | 55.9 | 8.7 |

Unlike MM-Verify, which generates $N$ responses in parallel and selects the top $k$ verified ones for majority voting, we adopt *Sherlock*'s sequential self-correction during inference. After each response, a verifier checks correctness; if incorrect, MM-Verify's critique guides the next round. Sampling stops once a correct response is verified. We compare this approach with parallel majority voting on MathVista [21], reporting accuracy and inference time (A100 GPU hours) in Table 5. Results show that a strong verifier improves both accuracy and efficiency. *Sherlock*'s self-correction achieves higher accuracy with lower cost, demonstrating superior scalability for complex reasoning tasks.

## 6 Conclusion and Discussion

In this paper, we introduce *Sherlock*, the first framework to achieve intrinsic self-correction in reasoning VLMs, with significant improvements across diverse benchmarks. Our analysis reveals that existing reasoning VLMs, whether trained with SFT or RL, struggle to self-correct. *Sherlock* addresses this gap with a novel trajectory-level objective, enabling self-correction and self-improvement using only 20k annotated examples. Our findings show that self-correction is not only feasible but also a powerful strategy to improve reasoning and inference-time scaling in VLMs. This makes VLMs more suitable for complex tasks, especially those where generating a correct answer in a single pass is difficult. Looking ahead, *Sherlock* provides a promising approach for extending self-correction to other types of reasoning models. It can also be further enhanced by incorporating step-wise self-correction for more efficient self-correction.

## Acknowledgements

We thank Xingchao Liu and the anonymous reviewers for their thoughtful comments on the manuscript. This research is supported in part by NSF IIS-2508145, Amazon Research Award, and OpenAI Researcher Access Program.

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

# A  Mathematical Derivations

In this section, we provide a detailed derivation of our *Sherlock* objective leading to Eq. 6. We extend the derivation structure of prior methods such as SRPO [5] and ARIES [52] from the response level to a more fine-grained trajectory level.

$$\max_{\pi} \mathbb{E}_{Y^2_{\geq i} \sim \pi(\cdot|[x_{I\&T}, Y^1, t; Y^1_{<i}])} \left[ p(Y^2_{\geq i} \succ Y^1_{\geq i}|x_{I\&T}; Y^1_{<i}) - \beta D_{\mathrm{KL}}(\pi \| \pi_{\mathrm{ref}}|[x_{I\&T}, Y^1, t; Y^1_{<i}]) \right]$$

$$+ \mathbb{E}_{Y^2_{\geq i} \sim \pi(\cdot|[x_{I\&T}, Y^1, t; Y^2_{<i}])} \left[ p(Y^2_{\geq i} \succ Y^1_{\geq i}|x_{I\&T}; Y^2_{<i}) - \beta D_{\mathrm{KL}}(\pi \| \pi_{\mathrm{ref}}|[x_{I\&T}, Y^1, t; Y^2_{<i}]) \right] \quad (9)$$

$$= \max_{\pi} \mathbb{E}_{Y^2_{\geq i} \sim \pi(\cdot|[x_{I\&T}, Y^1, t; Y^2_{<i}])} \left[ p(Y^2_{\geq i} \succ Y^1_{\geq i}|x_{I\&T}; Y^2_{<i}) - \beta \log \frac{\pi(Y^2_{\geq i}|[x_{I\&T}, Y^1, t; Y^2_{<i}])}{\pi_{\mathrm{ref}}(Y^2_{\geq i}|[x_{I\&T}, Y^1, t; Y^2_{<i}])} \right]$$

$$+ \mathbb{E}_{Y^2_{\geq i} \sim \pi(\cdot|[x_{I\&T}, Y^1, t; Y^1_{<i}])} \left[ p(Y^2_{\geq i} \succ Y^1_{\geq i}|x_{I\&T}; Y^1_{<i}) - \beta \log \frac{\pi(Y^2_{\geq i}|[x_{I\&T}, Y^1, t; Y^1_{<i}])}{\pi_{\mathrm{ref}}(Y^2_{\geq i}|[x_{I\&T}, Y^1, t; Y^1_{<i}])} \right]$$

$$(10)$$

$$= \max_{\pi} \beta \mathbb{E}_{Y^2_{\geq i} \sim \pi(\cdot|[x_{I\&T}, Y^1, t; Y^2_{<i}])} \left[ \log \exp \left( \frac{p(Y^2_{\geq i} \succ Y^1_{\geq i}|x_{I\&T}; Y^2_{<i})}{\beta} \right) - \log \frac{\pi(Y^2_{\geq i}|[x_{I\&T}, Y^1, t; Y^2_{<i}])}{\pi_{\mathrm{ref}}(Y^2_{\geq i}|[x_{I\&T}, Y^1, t; Y^2_{<i}])} \right]$$

$$+ \beta \mathbb{E}_{Y^2_{\geq i} \sim \pi(\cdot|[x_{I\&T}, Y^1, t; Y^1_{<i}])} \left[ \log \exp \left( \frac{p(Y^2_{\geq i} \succ Y^1_{\geq i}|x_{I\&T}; Y^1_{<i})}{\beta} \right) - \log \frac{\pi(Y^2_{\geq i}|[x_{I\&T}, Y^1, t; Y^1_{<i}])}{\pi_{\mathrm{ref}}(Y^2_{\geq i}|[x_{I\&T}, Y^1, t; Y^1_{<i}])} \right]$$

$$(11)$$

$$= \max_{\pi} -\beta \mathbb{E}_{Y^2_{\geq i} \sim \pi(\cdot|[x_{I\&T}, Y^1, t; Y^2_{<i}])} \left[ \log \frac{\pi(Y^2_{\geq i}|[x_{I\&T}, Y^1, t; Y^2_{<i}]) Z(x_{I\&T}, Y^1, t; Y^2_{<i})}{\pi_{\mathrm{ref}}(Y^2_{\geq i}|[x_{I\&T}, Y^1, t; Y^2_{<i}]) \exp \left( \frac{p(Y^2_{\geq i} \succ Y^1_{\geq i}|x_{I\&T}; Y^2_{<i})}{\beta} \right)} \right]$$

$$- \beta \mathbb{E}_{Y^2_{\geq i} \sim \pi(\cdot|[x_{I\&T}, Y^1, t; Y^1_{<i}])} \left[ \log \frac{\pi(Y^2_{\geq i}|[x_{I\&T}, Y^1, t; Y^1_{<i}]) Z(x_{I\&T}, Y^1, t; Y^1_{<i})}{\pi_{\mathrm{ref}}(Y^2_{\geq i}|[x_{I\&T}, Y^1, t; Y^1_{<i}]) \exp \left( \frac{p(Y^2_{\geq i} \succ Y^1_{\geq i}|x_{I\&T}; Y^1_{<i})}{\beta} \right)} \right]$$

$$+ \beta \log Z(x_{I\&T}, Y^1, t; Y^2_{<i}) + \beta \log Z(x_{I\&T}, Y^1, t; Y^1_{<i}) \quad (12)$$

$$= \max_{\pi} -\beta D_{KL} \left( \pi(Y^2_{\geq i}|[x_{I\&T}, Y^1, t; Y^2_{<i}]) \,\middle\|\, \frac{\pi_{\mathrm{ref}}(Y^2_{\geq i}|[x_{I\&T}, Y^1, t; Y^2_{<i}]) \exp \left( \frac{p(Y^2_{\geq i} \succ Y^1_{\geq i}|x_{I\&T}; Y^2_{<i})}{\beta} \right)}{Z(x_{I\&T}, Y^1, t; Y^2_{<i})} \right)$$

$$- \beta D_{KL} \left( \pi(Y^2_{\geq i}|[x_{I\&T}, Y^1, t; Y^1_{<i}]) \,\middle\|\, \frac{\pi_{\mathrm{ref}}(Y^2_{\geq i}|[x_{I\&T}, Y^1, t; Y^2_{<i}]) \exp \left( \frac{p(Y^2_{\geq i} \succ Y^1_{\geq i}|x_{I\&T}; Y^1_{<i})}{\beta} \right)}{Z(x_{I\&T}, Y^1, t; Y^1_{<i})} \right)$$

$$+ \beta \log Z(x_{I\&T}, Y^1, t; Y^2_{<i}) + \beta \log Z(x_{I\&T}, Y^1, t; Y^1_{<i}), \quad (13)$$

where $Z(X, Y^1, z; Y^2_{<i})$ and $Z(X, Y^1, z; Y^1_{<i})$ denote the partition functions. Noting the non-negativity of the KL divergence, the optimal proxy is given by:

$$\pi^*(Y^2_{\geq i}|[x_{I\&T}, Y^1, t; Y^2_{<i}]) = \frac{\pi_{\mathrm{ref}}(Y^2_{\geq i}|[x_{I\&T}, Y^1, t; Y^2_{<i}]) \exp \left( \frac{p(Y^2_{\geq i} \succ Y^1_{\geq i}|x_{I\&T}; Y^2_{<i})}{\beta} \right)}{Z(x_{I\&T}, Y^1, t; Y^2_{<i})} \quad (14)$$

$$\pi^*(Y^2_{\geq i}|[x_{I\&T}, Y^1, t; Y^1_{<i}]) = \frac{\pi_{\mathrm{ref}}(Y^2_{\geq i}|[x_{I\&T}, Y^1, t; Y^1_{<i}]) \exp \left( \frac{p(Y^2_{\geq i} \succ Y^1_{\geq i}|x_{I\&T}; Y^1_{<i})}{\beta} \right)}{Z(x_{I\&T}, Y^1, t; Y^1_{<i})}. \quad (15)$$

Therefore, we have:

$$p(Y^2_{\geq i} \succ Y^1_{\geq i}|x_{I\&T}; Y^2_{<i}) = \beta \log \frac{\pi^*(Y^2_{\geq i}|[x_{I\&T}, Y^1, t; Y^2_{<i}])}{\pi_{\mathrm{ref}}(Y^2_{\geq i}|[x_{I\&T}, Y^1, t; Y^2_{<i}])} + \beta \log Z(x_{I\&T}, Y^1, t; Y^2_{<i}) \quad (16)$$

$$p(Y^2_{\geq i} \succ Y^1_{\geq i}|x_{I\&T}; Y^1_{<i}) = \beta \log \frac{\pi^*(Y^2_{\geq i}|[x_{I\&T}, Y^1, t; Y^1_{<i}])}{\pi_{\mathrm{ref}}(Y^2_{\geq i}|[x_{I\&T}, Y^1, t; Y^1_{<i}])} + \beta \log Z(x_{I\&T}, Y^1, t; Y^1_{<i}) \quad (17)$$

Recalling the definition of $p(Y^2_{\geq i} \succ Y^1_{\geq i}|x_{I\&T})$ in Eq. 5, we have $p(Y^1_{\geq i} \succ Y^1_{\geq i}|x_{I\&T}; Y^2_{<i}) = \frac{1}{2}$, and $p(Y^1_{\geq i} \succ Y^1_{\geq i}|x_{I\&T}; Y^1_{<i}) = \frac{1}{2}$. Then, combining it with Eq. 16, and 17, we obtain:

$$\frac{1}{2} = \beta \log \frac{\pi^*(Y^1_{\geq i}|[x_{I\&T}, Y^1, t; Y^1_{<i}])}{\pi_{\text{ref}}(Y^1_{\geq i}|[x_{I\&T}, Y^1, t; Y^1_{<i}])} + \beta \log Z(x_{I\&T}, Y^1, t; Y^1_{<i}) \tag{18}$$

$$\frac{1}{2} = \beta \log \frac{\pi^*(Y^1_{\geq i}|[x_{I\&T}, Y^1, t; Y^2_{<i}])}{\pi_{\text{ref}}(Y^1_{\geq i}|[x_{I\&T}, Y^1, t; Y^2_{<i}])} + \beta \log Z(x_{I\&T}, Y^1, t; Y^2_{<i}) \tag{19}$$

By subtracting Eq. 18 from Eq. 16, and Eq. 19 from Eq. 17, we obtain:

$$p(Y^2_{\geq i} \succ Y^1_{\geq i}|x_{I\&T}; Y^2_{<i}) - \frac{1}{2} = \beta \left( \log \frac{\pi^*(Y^2_{\geq i}|[x_{I\&T}, Y^1, t; Y^2_{<i}])}{\pi_{\text{ref}}(Y^2_{\geq i}|[x_{I\&T}, Y^1, t; Y^2_{<i}])} - \log \frac{\pi^*(Y^1_{\geq i}|[x_{I\&T}, Y^1, t; Y^1_{<i}])}{\pi_{\text{ref}}(Y^1_{\geq i}|[x_{I\&T}, Y^1, t; Y^1_{<i}])} \right)$$

$$+ \beta(\log Z(x_{I\&T}, Y^1, t; Y^2_{<i}) - \log Z(x_{I\&T}, Y^1, t; Y^1_{<i}))$$

$$= \beta \log \left( \frac{\pi^*(Y^2_{\geq i}|[x_{I\&T}, Y^1, t; Y^2_{<i}])\pi_{\text{ref}}(Y^1_{\geq i}|[x_{I\&T}, Y^1, t; Y^1_{<i}])}{\pi_{\text{ref}}(Y^2_{\geq i}|[x_{I\&T}, Y^1, t; Y^2_{<i}])\pi^*(Y^1_{\geq i}|[x_{I\&T}, Y^1, t; Y^1_{<i}])} \right) + \beta \log \left( \frac{Z(x_{I\&T}, Y^1, t; Y^2_{<i})}{Z(x_{I\&T}, Y^1, t; Y^1_{<i})} \right) \tag{20}$$

$$p(Y^2_{\geq i} \succ Y^1_{\geq i}|x_{I\&T}; Y^1_{<i}) - \frac{1}{2} = \beta \left( \log \frac{\pi^*(Y^2_{\geq i}|[x_{I\&T}, Y^1, t; Y^1_{<i}])}{\pi_{\text{ref}}(Y^2_{\geq i}|[x_{I\&T}, Y^1, t; Y^1_{<i}])} - \log \frac{\pi^*(Y^1_{\geq i}|[x_{I\&T}, Y^1, t; Y^2_{<i}])}{\pi_{\text{ref}}(Y^1_{\geq i}|[x_{I\&T}, Y^1, t; Y^2_{<i}])} \right)$$

$$+ \beta(\log Z(x_{I\&T}, Y^1, t; Y^1_{<i}) - \log Z(x_{I\&T}, Y^1, t; Y^2_{<i}))$$

$$= \beta \log \left( \frac{\pi^*(Y^2_{\geq i}|[x_{I\&T}, Y^1, t; Y^1_{<i}])\pi_{\text{ref}}(Y^1_{\geq i}|[x_{I\&T}, Y^1, t; Y^2_{<i}])}{\pi_{\text{ref}}(Y^2_{\geq i}|[x_{I\&T}, Y^1, t; Y^1_{<i}])\pi^*(Y^1_{\geq i}|[x_{I\&T}, Y^1, t; Y^2_{<i}])} \right) + \beta \log \left( \frac{Z(x_{I\&T}, Y^1, t; Y^1_{<i})}{Z(x_{I\&T}, Y^1, t; Y^2_{<i})} \right) \tag{21}$$

Next, by adding Eq. 20 and Eq. 21, we obtain the following expression:

$$v(x, Y^1, t; Y^1_{<i}, Y^2_{<i}; \pi_\theta) = \beta \log \frac{\pi_\theta(Y^2_{\geq i}|[x, Y^1, t; Y^2_{<i}])}{\pi_{\text{ref}}(Y^2_{\geq i}|[x, Y^1, t; Y^2_{<i}])} - \beta \log \frac{\pi_\theta(Y^1_{\geq i}|[x, Y^1, t; Y^1_{<i}])}{\pi_{\text{ref}}(Y^1_{\geq i}|[x, Y^1, t; Y^1_{<i}])}$$

$$u(x, Y^1, t; Y^1_{<i}, Y^2_{<i}; \pi_\theta) = \beta \log \frac{\pi_\theta(Y^2_{\geq i}|[x, Y^1, t; Y^1_{<i}])}{\pi_{\text{ref}}(Y^2_{\geq i}|[x, Y^1, t; Y^1_{<i}])} - \beta \log \frac{\pi_\theta(Y^1_{\geq i}|[x, Y^1, t; Y^2_{<i}])}{\pi_{\text{ref}}(Y^1_{\geq i}|[x, Y^1, t; Y^2_{<i}])}$$

$$p(Y^2_{\geq i} \succ Y^1_{\geq i}|x_{I\&T}; Y^2_{<i}) + p(Y^2_{\geq i} \succ Y^1_{\geq i}|x_{I\&T}; Y^1_{<i}) - 1$$

$$= v(x_{I\&T}, Y^1, t; Y^1_{<i}, Y^2_{<i}) + u(x_{I\&T}, Y^1, t; Y^1_{<i}, Y^2_{<i}) \tag{22}$$

Finally, we follow prior work [52, 25, 1] to adopt the Mean Squared Error (MSE) as the loss function and parameterize the policy model as $\pi_\theta$. The training is then performed on the preference dataset $\mathcal{D}$ using the following loss:

$$v(x, Y^l, t; Y^l_{<i}, Y^w_{<i}; \pi_\theta) = \beta \log \underbrace{\frac{\pi_\theta(Y^w_{\geq i}|[x, Y^l, t; Y^w_{<i}])}{\pi_{\text{ref}}(Y^w_{\geq i}|[x, Y^l, t; Y^w_{<i}])}}_{\text{Encourage positive self-correction}} - \beta \log \underbrace{\frac{\pi_\theta(Y^l_{\geq i}|[x, Y^l, t; Y^l_{<i}])}{\pi_{\text{ref}}(Y^l_{\geq i}|[x, Y^l, t; Y^l_{<i}])}}_{\text{Discourage no self-correction}}$$

$$u(x, Y^l, t; Y^l_{<i}, Y^w_{<i}; \pi_\theta) = \beta \log \underbrace{\frac{\pi_\theta(Y^w_{\geq i}|[x, Y^l, t; Y^l_{<i}])}{\pi_{\text{ref}}(Y^w_{\geq i}|[x, Y^l, t; Y^l_{<i}])}}_{\text{Encourage positive self-correction}} - \beta \log \underbrace{\frac{\pi_\theta(Y^l_{\geq i}|[x, Y^l, t; Y^w_{<i}])}{\pi_{\text{ref}}(Y^l_{\geq i}|[x, Y^l, t; Y^w_{<i}])}}_{\text{Discourage no self-correction}}$$

$$v(x, Y^w, t; Y^w_{<i}, Y^l_{<i}; \pi_\theta) = \beta \log \underbrace{\frac{\pi_\theta(Y^l_{\geq i}|[x, Y^w, t; Y^l_{<i}])}{\pi_{\text{ref}}(Y^l_{\geq i}|[x, Y^w, t; Y^l_{<i}])}}_{\text{Discourage negative self-correction}} - \beta \log \underbrace{\frac{\pi_\theta(Y^w_{\geq i}|[x, Y^w, t; Y^w_{<i}])}{\pi_{\text{ref}}(Y^w_{\geq i}|[x, Y^w, t; Y^w_{<i}])}}_{\text{Encourage no self-correction}}$$

$$u(x, Y^w, t; Y^w_{<i}, Y^l_{<i}; \pi_\theta) = \beta \log \underbrace{\frac{\pi_\theta(Y^l_{\geq i}|[x, Y^w, t; Y^w_{<i}])}{\pi_{\text{ref}}(Y^l_{\geq i}|[x, Y^w, t; Y^w_{<i}])}}_{\text{Discourage negative self-correction}} - \beta \log \underbrace{\frac{\pi_\theta(Y^w_{\geq i}|[x, Y^w, t; Y^l_{<i}])}{\pi_{\text{ref}}(Y^w_{\geq i}|[x, Y^w, t; Y^l_{<i}])}}_{\text{Encourage no self-correction}}$$

$$\mathcal{L}_{\text{SC}}(\pi_\theta; \pi_{\text{ref}}) = \mathbb{E}_{(x,Y^w,Y^l)\sim\mathcal{D}} \left[ 1 - v(x_{I\&T}, Y^l, t; Y^l_{<i}, Y^w_{<i}; \pi_\theta) - u(x_{I\&T}, Y^l, t; Y^l_{<i}, Y^w_{<i}; \pi_\theta) \right]^2$$
$$+ \mathbb{E}_{(x,Y^w,Y^l)\sim\mathcal{D}} \left[ 1 + v(x_{I\&T}, Y^w, t; Y^w_{<i}, Y^l_{<i}; \pi_\theta) + u(x_{I\&T}, Y^w, t; Y^w_{<i}, Y^l_{<i}; \pi_\theta) \right]^2. \quad (23)$$

Our self-correction loss is designed to primarily encourage the model to learn desirable self-correction behavior. Specifically, it promotes *positive self-correction*: revising a low-quality response $Y^l$ into a high-quality one $Y^w$, and supports *no self-correction* when the initial response is already correct, i.e., from $Y^w$ to $Y^w$. Conversely, the loss penalizes *negative self-correction*, such as degrading a good response $Y^w$ into a poor one $Y^l$, as well as failing to revise an incorrect response, i.e., $Y^l$ to $Y^l$.

# B  Implementation Details

## B.1  Self-Correction Prompt

Considering the four-stage thinking structure of the LLaVA-CoT [44] dataset, we design the following self-correction template by modifying the template proposed in ARIES [52].

---

**Self-Correction Prompt t**

Below is a QUESTION from a user and an EXAMPLE RESPONSE.
Please provide a more helpful RESPONSE, improving the EXAMPLE RESPONSE by making the content even clearer, more accurate, and with a reasonable logic. Focus on addressing the human's QUESTION step by step based on the image without including irrelevant content.

QUESTION:
{question}

EXAMPLE RESPONSE:
{example_response}

Now, refine and improve the RESPONSE further. You can consider two approaches:
1. REFINEMENT: If the SUMMARY section in the response is closely related to the question, the CAPTION section accurately describes the image, the REASONING section is logically clear and correct without any contradictions, and the CONCLUSION provides an accurate answer based on the previous steps, enhance clarity, accuracy, or reasoning logic as needed.
2. NEW RESPONSE: If the SUMMARY section incorrectly summarizes the intent of the issue, the CAPTION contains content unrelated to or incorrect about the image, there are logical errors or contradictions in the REASONING, or the CONCLUSION incorrectly states the findings, please enhance the accuracy and quality of each step, and craft a more effective RESPONSE that thoroughly resolves the QUESTION.

RESPONSE:

---

## B.2  Data Construction and Training Details

In this section, we present the detailed data construction procedures for each stage of the *Sherlock* framework. Specifically, a total of 20k annotated examples and 10k unlabeled questions are randomly sampled from the LLaVA-CoT [44] dataset for training.

**Stage I: SFT Cold-Starting.**  We randomly sample two sets of 10k annotated examples from the LLaVA-CoT dataset, denoted as $\mathcal{D}_A$ and $\mathcal{D}_B$. We first use $\mathcal{D}_A$ to train the Llama3.2-Vision-11B-Instruct [9] model for one epoch using vanilla SFT loss as follows, enabling the model to generate responses in a fixed reasoning template:

$$\mathcal{L}_{\text{SFT}}(\pi) = -\mathbb{E}_{(x_{I\&T}, Y)\sim\mathcal{D}_A}[\log \pi(Y|x_{I\&T})] \quad (24)$$

We denote the resulting model as R0 VLM. Next, we sample responses on $\mathcal{D}_B$ using R0 VLM, i.e., $Y^w \sim \pi_{\text{R0 VLM}}(\cdot \mid x_{I\&T})$, which yields relatively low-quality outputs following the CoT template. Each sampled response is paired with its corresponding annotated answer and input question to

Table 6: Detailed training hyperparameters for each stage of the *Sherlock* model.

| *Sherlock* model | Learning Rate | Max Length | Batch Size | $\alpha$ | $\beta_{\text{DPO}}$ | Warm-Up Ratio | Epoch |
|---|---|---|---|---|---|---|---|
| *SFT* | 1e-6 | 4096 | 128 | - | - | 0.03 | 3 |
| *Offline* | 5e-6 | 4096 | 32 | 0.25 | 0.1 | 0.00 | 1 |
| *Iter1* | 5e-7 | 4096 | 32 | 0.25 | 0.1 | 0.00 | 1 |
| *Iter2* | 5e-7 | 4096 | 32 | 0.25 | 0.1 | 0.00 | 1 |

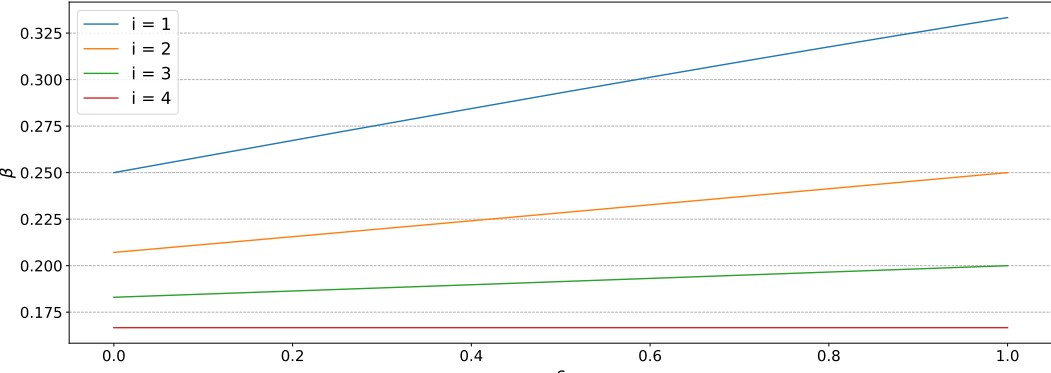

Figure 5: Values of dynamic $\beta$ under different truncation steps $i$ and visual perturbation levels $\epsilon$.

form the Sherlock SFT dataset, denoted as $\mathcal{D}_{\text{Sherlock-SFT}} = \{x, Y^l, Y^w\}$. Finally, we apply the Sherlock-SFT loss defined in Eq. 3 to cold-start the base VLM, jointly equipping it with both reasoning and self-correction capabilities.

**Stage II: Offline Training.** In the offline preference training stage, we construct trajectory-level preference data using the 10k examples in $\mathcal{D}_A$. The LLaVA-CoT dataset comprises four reasoning stages: `<SUMMARY>`, `<CAPTION>`, `<REASONING>`, and `<CONCLUSION>`, where the final `<CONCLUSION>` stage only provides the answer $a$. We randomly sample a truncation step $i \sim \mathcal{U}\{1, 2, 3, 4\}$ to determine the prefix. Specifically, when $i = 1$, no prefix is retained; when $i = 2, 3$, or $4$, we retain the SUMMARY, SUMMARY+CAPTION, and SUMMARY+CAPTION the first half of the REASONING stage, respectively. To generate a lower-quality suffix reasoning trajectory $Y^w_{\geq i} \sim \pi(\cdot | x_{I_\epsilon \& T}; Y_{<i})$, we introduce a random visual perturbation $\epsilon$ to the image input [30], which affects the model's visual perception during subsequent decoding.

**Stage III: Online Self-Improvement.** With the reasoning and self-correction capabilities of the *Sherlock Offline* model already activated, we aim to leverage the natural preference relationship between responses before and after correction to self-construct a preference dataset for iterative self-improvement training. Specifically, for each randomly sampled input question, we perform direct reasoning followed by three rounds of self-correction, obtaining responses $Y^1, Y^2, Y^3$, and $Y^4$. To reduce noise in the preference pairs, we apply a self-consistency filtering strategy: we only retain $Y^4$ as the preferred response $Y^w$ if the final answers in $Y^2, Y^3$, and $Y^4$ are semantically consistent. To construct the rejected response $Y^l$, we use $Y^1$ as the base and follow the same perturbation-based procedure as in Stage 2. It is worth noting that, in Stage 2, the chosen and rejected responses share the same prefix by construction, i.e., $Y^w_{<i} = Y^l_{<i}$. However, in Stage 3, since $Y^4$ is generated via three turn self-correction based on $Y^1$, their prefixes often differ, and thus the prefix portions of the preference pairs $Y^w_{<i}$ and $Y^l_{<i}$ are typically not equal.

**Dynamic $\beta$.** We build upon a static baseline of $\beta = 0.25$ and introduce controlled dynamism through randomly sampled truncation steps $i$ and image perturbation levels $\epsilon$. Considering that as reasoning progresses, the model's reliance on visual input typically decreases with longer generations, we aim to make $\beta$ less sensitive to changes in $\epsilon$ when $i$ is large. In addition, we aim to adopt more conservative policy updates when the quality gap between preference pairs is large, that is, when the truncation step $i$ is small and the image perturbation $\epsilon$ is large, while encouraging more assertive learning when the quality gap is small. Therefore, $\beta$ should be negatively correlated with $i$ and positively correlated with $\epsilon$. In Fig. 5, we visualize how $\beta$ varies with $\epsilon$ for $i = 1, 2, 3, 4$, demonstrating that the dynamic $\beta$ design in Eq. 8 aligns with our preferred properties.

**Training Details.** Our *Sherlock* is built on Llama3.2-Vision-11B-Instruct [35] model. In both the SFT and offline training stages, we randomly sample 10k annotated examples from the LLaVA-CoT [44] dataset, and follow the procedures introduced above to train the *Sherlock SFT* and *Offline* models with Eq. 3 and Eq. 7, respectively. We then perform two rounds of online self-improvement using 5k unlabeled data per round, containing only input questions and images. Following the self-consistency selection rule in Sec. 4.3, we construct a self-generated preference dataset. We then train the models using the objective in Eq. 7, resulting in *Sherlock Iter1* and *Iter2*. The detailed training hyperparameter is provided in Table 6.

**Sherlock Algorithm.** To provide a more intuitive understanding of our *Sherlock*, we summarize the detailed algorithm in Algorithm 1, including the training data, loss functions, and learning procedures.

---

**Algorithm 1:** *Sherlock*: Self-correction and Self-improvement training framework

---

**Input:** $\mathcal{D}_A = (x_{I\&T}, Y^w), \mathcal{D}_B = (x_{I\&T}, Y^w)$, base VLM $\pi$, iteration $T$, correction prompt $t$
**Output:** *Sherlock* model $\pi_{\text{Sherlock}}$

1 **Stage 1: SFT Cold-Start**
2    $\pi_{\text{VLM-R}} \leftarrow \arg\min_{\theta} \mathcal{L}_{\text{SFT}}(\pi_\theta)$ (Eq. 24) on $\mathcal{D}_A$
3    $\mathcal{D}_{\text{Sherlock-SFT}} \leftarrow \{(x_{I\&T}, Y^l, Y^w) | Y^l \sim \pi_{\text{R0 VLM}}(\cdot|x_{I\&T}), (x_{I\&T}, Y^w) \in \mathcal{D}_B\}$
4    $\pi_{\text{Sherlock-SFT}} \leftarrow \arg\min_{\theta} \mathcal{L}_{\text{Sherlock-SFT}}(\pi_\theta)$ (Eq. 3) on $\mathcal{D}_{\text{Sherlock-SFT}}$

5 **Stage 2: Offline Preference Training**
6    $\mathcal{D}_{\text{offline}} \leftarrow \emptyset$ **for** $(x_{I\&T}, Y^w)$ *in* $\mathcal{D}_A$ **do**
7       $i \sim \mathcal{U}\{0,1,2,3\}, \epsilon \sim \mathcal{U}(0,1)$
8       $Y^l_{<i} \leftarrow Y^w_{<i}, Y^l_{\geq i} \leftarrow \pi(\cdot|x_{I_\epsilon\&T}; Y^w_{\geq i}), \beta \leftarrow \beta(i,4,\epsilon)$ (Eq. 8)
9       $\mathcal{D}_{\text{offline}} \leftarrow \mathcal{D}_{\text{offline}} \cup \{x_{I\&T}, Y^l, Y^w, \beta\}$
10    $\pi_{\text{Sherlock-Offline}} \leftarrow \arg\min_{\theta} \mathcal{L}_{\text{Sherlock}}(\pi_\theta; \pi_{\text{ref}})$ (Eq. 7) on $\mathcal{D}_{\text{Offline}}$

11 **Stage 3: Online Iterative Self-Improvement**
12    $\pi_{\text{Sherlock-Iter 0}} \leftarrow \pi_{\text{Sherlock-Offline}}$
13 **for** $t = 1$ **to** $T$ **do**
14       $\mathcal{D}_{\text{Iter } t} \leftarrow \emptyset$
15       **while** $|\mathcal{D}_{\text{Iter } t}| < 5000$ **do**
16           Randomly sample question $x_{I\&T}$ without any annotation
17           $Y^1 \leftarrow \pi_{\text{Sherlock-Iter } t-1}(\cdot|x_{I\&T})$ **for** $j = 1$ **to** 3 **do**
18              $Y^{j+1} \leftarrow \pi_{\text{Sherlock-Iter } t-1}(\cdot|x_{I\&T}, Y^j, t)$
19           **if** $a^2 = a^3 = a^4$ **then**
20              $i \sim \mathcal{U}\{1,2,3,4\}, \epsilon \sim \mathcal{U}(0,1), Y^w \leftarrow Y^4, Y^l_{<i} \leftarrow Y^1_{<i}, \beta \leftarrow \beta(i,4,\epsilon)$ (Eq. 8)
21              $Y^l_{\geq i} \leftarrow \pi_{\text{Sherlock-Iter } t-1}(\cdot|x_{I_\epsilon\&T}, Y^l_{<i})$
22              $\mathcal{D}_{\text{Iter } t} \leftarrow \mathcal{D}_{\text{Iter } t} \cup \{x_{I\&T}, Y^l, Y^w, \beta\}$
23           **else**
24              Skip
25       $\pi_{\text{Sherlock-Iter } t} \leftarrow \arg\min_{\theta} \mathcal{L}_{\text{Sherlock}}(\pi_\theta; \pi_{\text{ref}})$ (Eq. 7) on $\mathcal{D}_{\text{Iter } t}$

---

## C   Evaluation Detail

### C.1   Evaluation Benchmarks

We conduct experiments on eight multimodal tasks, covering comprehensive VQA benchmarks (MMBench-V1.1 [20], MMVet [50], MME [17], MMStar [4]), math and science benchmarks (Math-Vista [21], AI2D [15], MMMU [51]), and a hallucination benchmark (HallusionBench [10]). We evaluate *Sherlock* under two settings: direct generation and after three rounds of self-correction. For consistency and reproducibility, all models are evaluated using the VLMEvalKit [8] pipeline.

**MMBench-V1.1.**    MMBench-v1.1 contains a total of 3217 samples and is designed to evaluate the visual perception and visual reasoning capabilities of VLMs. It adopts the CircularEval protocol, which presents each question to a VLM multiple times with shuffled answer choices. A model is considered successful only if it selects the correct answer in all attempts, thereby reducing the influence of random guessing in multiple-choice questions.

**MMVet.**    MMVet consists of 218 open-ended questions designed to comprehensively assess VLMs across six dimensions: Recognition, OCR, Knowledge, Language Generation, Spatial Awareness, and Math. The final evaluation results are obtained by scoring the model-generated open-ended responses using `GPT4-Turbo` as the grader.

**MME.**    MME is a comprehensive multimodal benchmark designed to assess two core capabilities of VLMs: visual perception and reasoning. It consists of 10 perception tasks and 4 reasoning tasks. The overall perception and reasoning (cognition) scores are computed as the sum of their respective subtask scores, with the total maximum score across all tasks being 2800.

**MMStar.**    MMStar consists of 1500 test samples and aims to address key issues in evaluation, such as low visual-textual alignment and potential data leakage during training. The benchmark is carefully curated and covers 6 core capabilities and 18 detailed evaluation axes.

**MathVista.**    MathVista introduces a unified benchmark with 6141 examples collected from 28 existing datasets and 3 newly curated ones (IQTest, FunctionQA, and PaperQA). In our experiments, we evaluate models on the `test-mini` split, which contains 1000 samples.

**MMMU.**    MMMU is an expert-level multimodal benchmark developed to assess the perception, knowledge, and reasoning abilities of VLMs. For our experiments, we use its validation set, which comprises 900 multimodal samples.

**AI2D.**    We evaluate model performance on the `AI2D_TEST_NO_MASK` setting from VLMEvalKit, which assesses VLMs' ability to interpret grade-school science diagrams.

**HallusionBench.**    HallusionBench is a challenging benchmark designed to evaluate image-context reasoning in VLMs. It consists of 346 images and 1129 expert-crafted questions, targeting nuanced visual understanding. We report the results as the average of the three evaluation metrics provided by VLMEvalKit.

## C.2    Evaluation Baselines

Considering that our approach is primarily initialized using the reasoning data from LLaVA-CoT [44], our main experiments compare with other SFT-based methods built on the Llama3.2-Vision-11B-Instruct [9] model, including LLaVA-CoT, Mulberry [48], and LlamaV-o1 [35]. In addition, to further investigate the self-correction behavior of reasoning VLMs, we also include experiments and analysis on the RL-based method VL-Rethinker [37].

**LLaVA-CoT.**    LLaVA-CoT constructs 100k stage-level Chain-of-Thought (CoT) annotations using multi-turn prompts to GPT-4o, and then trains the model via standard supervised fine-tuning (SFT) to acquire reasoning ability. Each response is structured into four stages: `<SUMMARY>`, `<CAPTION>`, `<REASONING>`, and `<CONCLUSION>`.

**Mulberry.**    Mulberry constructs 260k long CoT annotations by applying Monte Carlo Tree Search (MCTS) over strong models such as Qwen2-VL and GPT-4o, with a particular focus on the mathematical domain. In addition, it incorporates supplementary self-reflection SFT data to encourage models to engage in step-wise reasoning, first reflecting on prior errors and then performing corrections.

**LlamaV-o1.**    LlamaV-o1 extends the LLaVA-CoT dataset to 175k samples and adopts a multi-turn questioning and training strategy during inference to simulate the four reasoning stages: summary, caption, reasoning, and conclusion, thereby enabling step-by-step reasoning behavior.

**VL-Rethinker.** VL-Rethinker is built upon the Qwen2.5-VL series models and employs GRPO-based reinforcement learning to enhance reasoning capabilities. Additionally, it introduces a limited amount of self-reflection and correction data to enforce a self-reflection objective, aiming to train the model to revise earlier responses within a single thinking process.

### C.3 Evaluation Setup in Section 3

**Prompt for Modify One Step Setting.** Below is the prompt specific to LLaVA-CoT models.

---

**Modify One Step Prompt for Qwen2.5-Instruct-7B on LLaVA-CoT**

You are a professional sentence rewriting expert. Your task is to modify the given sentence by altering key numbers, or key phrases so that the resulting sentence conveys a different or incorrect meaning, finally leads to wrong answers. In essence, you should transform a logically correct or factually accurate statement into one that contains a logical or mathematical error. You should output the original format but with incorrect logic or meaning thay may leads wrong answers. Ensure that the intention or logic of your version clearly differs from the original input.

<EXAMPLE>
[INPUT]
### Question
Find $RS$ if $\triangle QRS$ is an equilateral triangle.
### Answer
<SUMMARY> To solve the problem, I will use the properties of an equilateral triangle to find the length of side $RS$. I will set up equations based on the given side lengths and solve for $RS$. </SUMMARY>
<CAPTION> The image shows an equilateral triangle $\triangle QRS$ with side lengths labeled as $QR = 4x$, $QS = 6x - 1$, and $RS = 2x + 1$. </CAPTION>
<REASONING> The triangle is equilateral, so all sides are equal. Therefore, we can set up the equation $QR = QS = RS$. This gives us the equations $4x = 6x - 1$ and $4x = 2x + 1$.

[YOUR REWRITTEN]
<SUMMARY> To solve the problem, I will use the properties of an equilateral triangle to find the length of side $RS$. I will set up equations based on the given side lengths and solve for $RS$. </SUMMARY>
<CAPTION> The image shows an equilateral triangle $\triangle QRS$ with side lengths labeled as $QR = 4x$, $QS = 9x - 7$, and $RS = 2x - 1$. </CAPTION>
<REASONING> The triangle is equilateral, so all sides are equal. Therefore, we can set up the equation $QR = QS = RS$. This gives us the equations $4x = 9x - 7$ and $4x = 2x - 1$. Make sure only output the rewritten answer with original format.

[INPUT]
{input}

[YOUR REWRITTEN]
Make sure only output your rewritten answer with original format, not output the question.

RESPONSE:

---

Below is the prompt specific to VL-Rethinker models.

---

**Modify One Step Prompt for Qwen2.5-Instruct-7B on VL-Rethinker**

You are a professional sentence rewriting expert. Your task is to modify the given sentence by altering key numbers, or key phrases so that the resulting sentence conveys a different or incorrect meaning, finally leads to wrong answers. In essence, you should transform a logically correct or factually accurate statement into one that contains a logical or mathematical error. You should output the original format but with incorrect logic or meaning thay may leads wrong answers. Ensure that the intention or logic of your version clearly differs from the original input.

---

<EXAMPLE>
[INPUT]
### Question
Find $RS$ if $\triangle QRS$ is an equilateral triangle.
### Answer
Since triangle QRS is an equilateral triangle, all its sides are equal in length. Therefore, we can set the expressions for the sides equal to each other and solve for x. The expressions for the sides are: - QR = 4x - RS = 2x + 1

[YOUR REWRITTEN]
Since triangle QRS is an equilateral triangle, all its sides are equal in length. Therefore, we can set the expressions for the sides equal to each other and solve for x. The expressions for the sides are: - QR = 6x - RS = 2x - 1
Make sure to make the rewritten sentence with explicit wrong steps or logic.

[INPUT]
{input}

[YOUR REWRITTEN]

**Prompt for External Critique.** In the external critique setting, we primarily follow the setup of Critic-V [53], which includes both the critique generation prompt and the correction prompt.

**External Critique Generation Prompt**

#### Question
{question}
#### Answer
{result}
#### Task
Please provide a critique of the answer above.

**Self-Correction Prompt via External Critique**

Reflection on former answer:
#### Former Answer
{answer}
{critics}
#### Question
{original_question}

## C.4 Evaluation Setup in Section 5

We report *Sherlock*'s performance under direct reasoning and three-round self-correction settings. We also evaluate MM-Verify as a verification mechanism to accelerate inference-time scaling, with the corresponding prompt shown below.

**Prompt for MM-Verify**

Solve the math problems and provide step-by-step solutions, ending with "The answer is [Insert Final Answer Here]".
When asked "Verification: Is the answer correct (Yes/No)?", respond with " Yes" or " No" based on the answer's correctness.
When asked "Verification: Let's verify step by step.", verify every step of the solution and conclude with "Verification: Is the answer correct (Yes/No)?" followed by " Yes" or " No".

Q:

{question}

A: Let's think step by step.

{answer}

Verification: Let's verify step by step.

---

**Self-Correction Prompt Based on Critique from MM-Verify**

Below is a QUESTION from a user and an EXAMPLE RESPONSE.
Please provide a more helpful RESPONSE, improving the EXAMPLE RESPONSE by making the content even clearer, more accurate, and with a reasonable logic.
Focus on addressing the human's QUESTION step by step based on the image without including irrelevant content.

QUESTION:
{question}

EXAMPLE RESPONSE:
{example_response}

CRITIQUE:
{critique}

Now, refine and improve the RESPONSE based on provided CRITIQUE. You can consider two approaches:
1. REFINEMENT: If the SUMMARY section in the response is closely related to the question, the CAPTION section accurately describes the image, the REASONING section is logically clear and correct without any contradictions, and the CONCLUSION provides an accurate answer based on the previous steps, enhance clarity, accuracy, or reasoning logic as needed.
2. NEW RESPONSE: If the SUMMARY section incorrectly summarizes the intent of the issue, the CAPTION contains content unrelated to or incorrect about the image, there are logical errors or contradictions in the REASONING, or the CONCLUSION incorrectly states the findings, please enhance the accuracy and quality of each step, and craft a more effective RESPONSE that thoroughly resolves the QUESTION.

RESPONSE:

# D Experimental Results

## D.1 Training Efficiency of Sherlock

Table 7: Training efficiency and performance compared to other baselines.

| Methods | Training Time (↓) | Accuracy (↑) |
|---|---|---|
| LLaVA-CoT [44] | 160 | 63.2 |
| LlamaV-o1 [35] | 288 | 63.4 |
| Sherlock | 128 | 64.1 |

Although *Sherlock* adopts a multi-stage framework, its data efficiency and RL-free design make training highly efficient. As shown in Table 7, *Sherlock* outperforms LLaVA-CoT and LlamaV-o1 while requiring the least training time.

## D.2 More Results of Sherlock Performance

**Sherlock Performance of Each Correction Turns.** We report the average performance of the *Sherlock Iter2* model across eight benchmarks under different numbers of self-correction rounds. As shown in Fig. 6, the *Sherlock* model consistently improves its accuracy with more rounds of self-correction, demonstrating stable inference-time scaling.

**Initialize Sherlock on Reasoning VLMs.** The goal of Sherlock is not merely to improve upon existing CoT models, but to introduce a new framework that transforms a base VLM into a reasoning

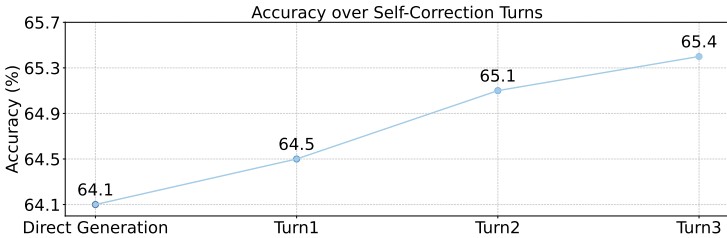

Figure 6: Average accuracy of the *Sherlock Iter2* model across eight benchmarks under direct generation and varying numbers of self-correction turns.

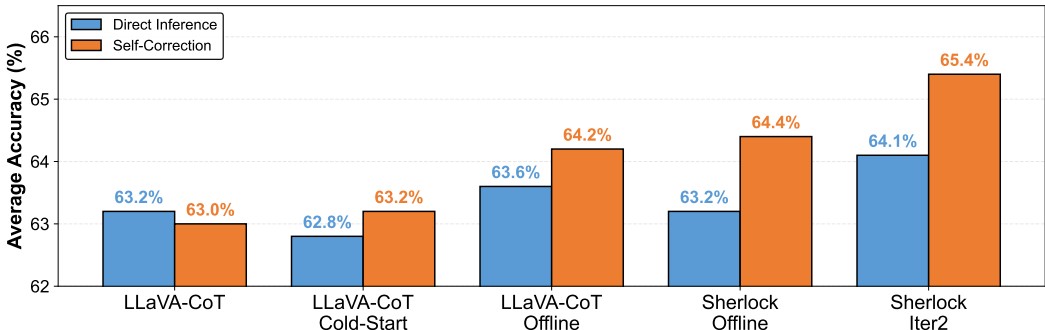

Figure 7: Performance comparison of different initialization strategies. LLaVA-CoT Cold-Start and LLaVA-CoT Offline denote post-training the LLaVA-CoT model using the "Cold-Start" and "Offline" stage strategies of Sherlock, respectively.

VLM with strong reasoning and self-correction capabilities. However, if we use LLaVA-CoT as the base VLM for Sherlock's multi-stage training, severe overfitting may occur because Sherlock's training data are randomly sampled from LLaVA-CoT 100k, meaning the model has already extensively seen these samples during its original training. To address this issue, we instead consider applying the offline stage's preference training objective to post fine-tune the LLaVA-CoT model directly. Results in Fig. 7 reveal two key observations. First, initializing LLaVA-CoT with the Cold-Start stage enhances the model's self-correction ability but slightly degrades its direct inference accuracy, which is consistent with our previous analysis. Second, applying offline preference training on LLaVA-CoT successfully improves both direct generation and self-correction capabilities. However, compared to Sherlock Offline, the self-correction ability remains limited, as the training data have already been seen during LLaVA-CoT's SFT stage.

Table 8: Performance comparison between LLaVA-Reasoner-DPO and Sherlock across 8 benchmarks.

| Models | #Data w/ GT | MMB | MMVet | Hallus | MMMU | MMStar | AI2D | MathV | MME | Avg. |
|---|---|---|---|---|---|---|---|---|---|---|
| Llama3.2V-11B-Ins | - | 65.8 | 57.6 | 42.7 | 47.8 | 53.0 | 88.2 | 49.7 | 1822 | 58.7 |
| LLaVA-Reaoner-DPO | 258k | 73.5 | 56.2 | 45.9 | 42.4 | 57.1 | 87.6 | 50.8 | 2001 | 60.6 |
| + Self-Correction | | 75.5 | 53.9 | 43.1 | 43.9 | 54.6 | 87.8 | 46.0 | 1875 | 59.0[1.6↓] |
| *Sherlock Iter2* | 20k | 74.6 | 62.4 | 48.7 | 49.7 | 57.7 | 89.6 | 52.0 | 2197 | 64.1 |
| + Self-Correction | | 77.2 | 62.6 | 51.2 | 50.1 | 59.0 | 90.6 | 54.0 | 2204 | 65.4[1.3↑] |

**Comparison with More Baseline.** To further validate the effectiveness of Sherlock, we compared its performance with LLaVA-Reasoner-DPO [56] across eight benchmarks. LLaVA-Reasoner-DPO is trained via DPO [29] by distilling from a stronger model (GPT-4o) and constructing preference pairs based on answer accuracy. For a fair comparison, we trained LLaVA-Reasoner-DPO using Llama3.2-Vision-11B-Instruct [9] as the base model. The results in Table 8 show that, even with a stronger model initialization, LLaVA-Reasoner-DPO fails to further improve performance through CoT training, revealing its limitations. In contrast, Sherlock significantly enhances both the model's direct reasoning and self-correction capabilities.

# E   Case Study

---

### Case 1 of Sherlock Self-Correction

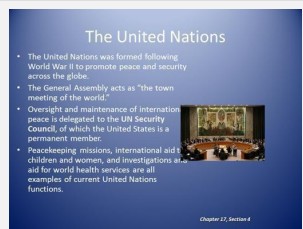

**Problem:**  What color is the text in the image?
(A) White (B) Blue (C) Black (D) Grey

**Sherlock Direct Generation Response.**

`<SUMMARY>` I will analyze the image to determine the color of the text by examining its appearance and comparing it to the given options. `</SUMMARY>`
`<CAPTION>` The image is a slide titled "The United Nations," containing bullet points about the organization's formation and functions. The text is set against a blue background, and there is an image of a UN General Assembly meeting. `</CAPTION>`
`<REASONING>` To determine the color of the text, I will look at its hue and brightness in contrast to the background. The text appears to be a lighter shade compared to the blue background, indicating it is a neutral color such as grey or white. Since it is not black, option C can be eliminated. Comparing it to the other options, white would be too bright, so it is likely grey. `</REASONING>`
`<CONCLUSION>` D ✗ `</CONCLUSION>`

**Sherlock Self-Correction Turn 1 Response.**

`<SUMMARY>` I will analyze the image to determine the color of the text by examining its appearance and comparing it to the given options. `</SUMMARY>`
`<CAPTION>` The image is a slide titled "The United Nations." It contains bullet points about the United Nations' formation and functions, with an accompanying image of a UN General Assembly meeting. The text is prominently displayed against a blue background. `</CAPTION>`
`<REASONING>` To determine the text color, I will look at its contrast with the background. The text appears lighter than the blue background, suggesting it is a neutral color. Comparing it to the options provided, the closest match is grey, as it typically appears lighter than black or blue and not as bright as white. `</REASONING>`
`<CONCLUSION>` D ✗ `</CONCLUSION>`

**Sherlock Self-Correction Turn 2 Response.**

`<SUMMARY>` To determine the color of the text in the image, I will carefully examine the image to identify and describe the text's color, then confirm it matches one of the given options. `</SUMMARY>`
`<CAPTION>` The image is a slide titled "The United Nations," containing information about the organization's formation and functions. It includes a photograph of a meeting at the United Nations General Assembly. The text is presented in a clear, legible font against a gradient blue background. `</CAPTION>`
`<REASONING>` Upon examining the image, the text stands out against the blue background. Its color appears to be a neutral shade that contrasts with the background, which is typically characteristic of the color gray. This observation aligns with the options provided, helping to identify the text color accurately. `</REASONING>`
`<CONCLUSION>` D ✗ `</CONCLUSION>`

**Sherlock Self-Correction Turn 3 Response.**

`<SUMMARY>` To determine the color of the text in the image, I will carefully analyze the visual elements and identify the color of the text by comparing it to the background and other elements in the image. `</SUMMARY>`
`<CAPTION>` The image is a slide titled "The United Nations," containing bullet points about the history and functions of the United Nations. The text is presented against a gradient blue background, with an image of a United Nations General Assembly session on the right. `</CAPTION>`
`<REASONING>` The text on the slide is clearly visible and stands out against the blue background. By comparing the text's color to the surrounding blue hues, I can determine that the text is a different color. The text appears to be a neutral, light color that contrasts with the darker shades of blue in the background. `</REASONING>`
`<CONCLUSION>` A ✓ `</CONCLUSION>`

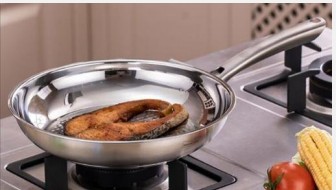

**Problem:** What's the function of the demonstrated object?
(A) Cooking (B) Cook soup (C) Fry (D) steam

**Sherlock Direct Generation Response.**

<SUMMARY> I will analyze the image to identify the object and its function, focusing on its features and context to determine the correct answer from the given options. </SUMMARY>
<CAPTION> The image shows a stainless steel pan on a stovetop. Inside the pan, there is a piece of fish being cooked in water. There are also vegetables, such as corn and a tomato, visible nearby on the stove. </CAPTION>
<REASONING> The object in the image is a stainless steel pan placed on a stove, containing water and a piece of fish. The presence of heat from the stove and the fish being submerged in water indicates that it is being used for cooking. Among the options provided, "Cooking" is the most general term that encompasses the activity taking place. While the pan could be used to fry, cook soup, or steam, the most straightforward answer based on the image alone is that it is being used for cooking in general. </REASONING>
<CONCLUSION> (A) Cooking ✗ </CONCLUSION>

**Sherlock Self-Correction Turn 1 Response.**

<SUMMARY> I will analyze the image to identify the object and its function, focusing on its features and context to determine the correct answer. </SUMMARY>
<CAPTION> The image shows a stainless steel frying pan on a gas stove. Inside the pan, there is a piece of fish being cooked in oil or melted fat, indicating that it is being used for cooking. There are also some vegetables, such as corn and a tomato, visible nearby. </CAPTION>
<REASONING> The object in the image is a frying pan, which is typically used for frying foods. The presence of the fish being cooked in the pan suggests that it is being used for frying. The pan's shape and the cooking environment further support this conclusion, as frying usually involves cooking in a pan over direct heat. </REASONING>
<CONCLUSION> (C) Fry ✓ </CONCLUSION>

**Sherlock Self-Correction Turn 2 Response.**

<SUMMARY> I will analyze the image to identify the object and its function, then determine which option best describes its use based on its features and context. </SUMMARY>
<CAPTION> The image shows a stainless steel pan on a stovetop, containing water with a piece of fish submerged. There are vegetables, such as corn and a tomato, nearby on the counter. </CAPTION>
<REASONING> The object is a stainless steel pan placed on a stove, indicating it is designed for cooking. It contains water and a piece of fish, which suggests that it is being used to cook the fish. The presence of the stovetop and the way the fish is submerged in water further implies that the pan is being used for a cooking method that involves boiling or frying. Given the options, frying typically involves submerging food in hot oil or water and is consistent with the setup shown. </REASONING>
<CONCLUSION> C ✓ </CONCLUSION>

**Sherlock Self-Correction Turn 3 Response.**

<SUMMARY> I will analyze the image to identify the object and its function, using visual cues to determine the correct answer from the given options. </SUMMARY>
<CAPTION> The image shows a stainless steel frying pan on a gas stove. Inside the pan, there is a piece of fish being cooked in oil or melted fat, indicating that it is being used for cooking. There are also some vegetables, such as corn and a tomato, visible in the background. </CAPTION>
<REASONING> The object in the image is a frying pan, which is typically used for frying foods. The presence of the fish cooking in the pan with a visible layer of oil further supports this function. Frying involves cooking food in hot oil or fat, which matches the scenario depicted. Other options like cooking soup or steaming require different types of vessels, such as pots with a lid for steaming or larger pots for soup. </REASONING>
<CONCLUSION> C ✓ </CONCLUSION>

## Case 3 of Sherlock Self-Correction

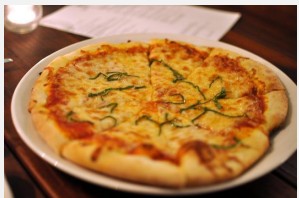

**Problem:** Hint: Please answer the question and provide the correct option letter, e.g., A, B, C, D, at the end.
Question: Is it split in half?
Choices: (A) Yes (B) No

**Sherlock Direct Generation Response.**

<SUMMARY> I will analyze the image to determine if the pizza is split in half by examining its shape and any visible cuts. I will then compare my findings with the given options to provide the correct answer. </SUMMARY>
<CAPTION> The image shows a pizza on a white plate. The pizza has a thin crust and is topped with cheese and basil. It appears to have been cut into several slices, with two main cuts extending from the center towards the edge, suggesting it is divided into halves. </CAPTION>
<REASONING> To determine if the pizza is split in half, I will look for clear cuts that divide the pizza into two main portions. The pizza has two prominent diagonal cuts starting from the center, meeting the edge and creating two large halves. Additional smaller cuts are also visible, but the primary division into two is clear. </REASONING>
<CONCLUSION> A ✗ </CONCLUSION>

**Sherlock Self-Correction Turn 1 Response.**

<SUMMARY> I will analyze the image to determine if the pizza is split in half by examining its cut pattern and describe my findings systematically to reach the correct conclusion. </SUMMARY>
<CAPTION> The image shows a pizza on a white plate. The pizza has a thin crust and is topped with cheese and basil strips. It is cut into several slices, with two prominent cuts that divide it into halves. </CAPTION>
<REASONING> To determine if the pizza is split in half, I will look at the number and pattern of the cuts. The pizza has two main cuts that run from the center to the edge, dividing it into two equal halves. There are additional smaller cuts, but the primary division is into two halves. </REASONING>
<CONCLUSION> A ✗ </CONCLUSION>

**Sherlock Self-Correction Turn 2 Response.**

<SUMMARY> I will analyze the image to determine if the pizza is split in half by examining the number of slices and their arrangement. I will then compare my findings with the given options to provide the correct answer. </SUMMARY>
<CAPTION> The image shows a pizza on a white plate. The pizza has a thin crust and is topped with cheese and basil. It appears to be cut into eight slices, arranged in a symmetrical pattern. </CAPTION>
<REASONING> To determine if the pizza is split in half, I will count the number of slices. If it has two large slices separated by a clear line down the middle, it would be considered split in half. However, the pizza is cut into eight slices, indicating it is not divided simply into two halves. The slices are evenly distributed across the pizza with no two large halves. </REASONING>
<CONCLUSION> B ✓ </CONCLUSION>

**Sherlock Self-Correction Turn 3 Response.**

<SUMMARY> I will analyze the image to determine if the pizza is split in half by examining the number of slices and their arrangement. I will follow a structured approach to reach the conclusion. </SUMMARY>
<CAPTION> The image shows a pizza on a white plate. The pizza has a thin crust with a golden-brown appearance, topped with melted cheese and garnished with thin strips of green herbs. It is cut into eight slices, with four slices extending from the center towards the edge, forming a radial pattern. </CAPTION>
<REASONING> To determine if the pizza is split in half, I will assess the number and arrangement of the slices. A pizza that is split in half would typically have two large halves separated by a clear line down the middle, with no other slices present. In this image, the pizza is cut into eight equal slices, arranged in a symmetrical pattern, but there are not two primary halves. Therefore, it is not split in half. </REASONING>
<CONCLUSION> B ✓ </CONCLUSION>

## Case 4 of Sherlock Self-Correction

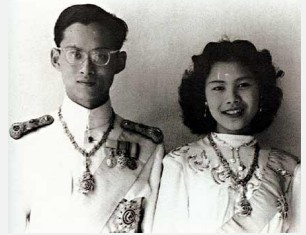

**Problem:** Hint: Please answer the question requiring an integer answer and provide the final value, e.g., 1, 2, 3, at the end.

Question: What is the age gap between these two people in image? (Unit: years)

**Sherlock Direct Generation Response.**

<SUMMARY> I will analyze the image to identify any visible clues that indicate the age difference between the two individuals, such as differences in their attire, accessories, or any other visible signs that might suggest a generational or age gap. </SUMMARY>

<CAPTION> The image shows a man and a woman dressed in traditional Thai military and royal attire, respectively. Both are adorned with medals and jewelry, suggesting a formal and ceremonial occasion. The woman's hairstyle and the style of their clothing further imply a historical context. </CAPTION>

<REASONING> To determine the age gap, I will consider the difference in their roles and the style of their clothing. The man's military uniform and the woman's royal attire suggest they are in a position of authority or royalty. The woman's youthful appearance and the style of her clothing may indicate she is younger than the man, consistent with a couple where the man is older. Given the royal context, it is plausible that the woman is his consort or wife, which often involves an age gap in royal marriages. </REASONING>

<CONCLUSION> 12 ✗ </CONCLUSION>

**Sherlock Self-Correction Turn 1 Response.**

<SUMMARY> To determine the age gap between the two individuals in the image, I will analyze historical context and available information about their birth dates. I will then calculate the difference to provide the answer. </SUMMARY>

<CAPTION> The image shows a man and a woman dressed in formal royal attire, both adorned with medals. The man is wearing a military uniform with epaulets, and the woman is in a formal, ornate dress. They are likely members of a royal family, given their attire and decorations. </CAPTION>

<REASONING> The individuals are King Bhumibol Adulyadej and Queen Sirikit of Thailand. King Bhumibol was born on December 5, 1927, and Queen Sirikit was born on August 12, 1932. To find the age gap, subtract the queen's birth year from the king's birth year: 1927 - 1932 = 5 years. </REASONING>

<CONCLUSION> 5 ✓ </CONCLUSION>

**Sherlock Self-Correction Turn 2 Response.**

<SUMMARY> To solve the problem, I will analyze the image to identify the individuals and determine the age gap between them. I will use historical information about their birth dates to calculate the age difference. </SUMMARY>

<CAPTION> The image shows a man and a woman dressed in formal royal attire, both adorned with medals. The man is wearing a military uniform, and the woman is in a matching ceremonial outfit. They are likely members of a royal family, given their clothing and decorations. </CAPTION>

<REASONING> The individuals in the image are King Bhumibol Adulyadej and Queen Sirikit of Thailand. King Bhumibol was born on December 5, 1927, and Queen Sirikit was born on August 12, 1932. To find the age gap, subtract the queen's birth year from the king's birth year: 1927 - 1932 = 5 years. </REASONING>

<CONCLUSION> 5 ✓ </CONCLUSION>

**Sherlock Self-Correction Turn 3 Response.**

<SUMMARY> To determine the age gap between the two individuals in the image, I will analyze historical data about their birth dates and calculate the difference. </SUMMARY>

<CAPTION> The image shows a man and a woman dressed in formal royal attire, both adorned with medals. They are likely members of a royal family, given their clothing and decorations. </CAPTION>

<REASONING> The man is King Bhumibol Adulyadej, and the woman is Queen Sirikit of Thailand. King Bhumibol was born on December 5, 1927, and Queen Sirikit was born on August 12, 1932. To find the age gap, subtract the queen's birth year from the king's birth year: 1927 - 1932 = 5 years. </REASONING>

<CONCLUSION> 5 ✓ </CONCLUSION>

