# OpenReview forum: "Sherlock: Self-Correcting Reasoning in Vision-Language Models"
_NeurIPS.cc/2025/Conference — NeurIPS 2025 poster_

### Official Review · Reviewer_sFen · 2025-06-20

**Clarity:** 3
**Significance:** 2
**Originality:** 3
**Rating:** 4
**Confidence:** 1

**Summary:**

The paper addresses the limitations of Vision-Language Models (VLMs) in reasoning tasks, specifically their sensitivity to errors, heavy dependence on annotated data, and limited generalization. It proposes a framework called Sherlock that leverages self-correction at the trajectory-level, incorporating visual perturbation-based preference data and dynamic β adjustment. Sherlock significantly reduces annotation requirements and enables continual self-improvement without external supervision, achieving superior performance compared to existing methods.

**Questions:**

Please see the weakness part.

**Ethical Concerns:**

["NO or VERY MINOR ethics concerns only"]

**Final Justification:**

The response has addressed most of my previous concerns and questions. However, as I am not very familiar with this area, I prefer to keep my original score.

**Limitations:**

yes

**Paper Formatting Concerns:**

1. There is not conclusion section in the paper.

**Quality:**

2

**Strengths And Weaknesses:**

Strength:

1.The paper clearly articulates the key limitations of existing reasoning VLMs—such as error propagation, data inefficiency, and poor generalization. The motivation for introducing self-correction is well-founded and compelling, particularly in the context of enabling more robust and scalable multimodal reasoning.


Weakness:

1. A major concern is the limited discussion of related work. While several components of Sherlock's algorithm appear to draw on techniques recently explored in the LLM literature, the paper lacks a clear delineation of which aspects are novel or uniquely adapted to the multimodal setting. It would greatly benefit from explicitly clarifying what parts of the method are specific to vision-language reasoning.

2. Sherlock's multi-stage training pipeline—comprising SFT, offline preference training, and iterative self-improvement—adds significant complexity. However, the paper provides limited discussion on how this design might be deployed or tuned in practice. Moreover, while an ablation study is provided, it does not clearly isolate the contribution of each individual loss component (e.g., DPO vs. self-correction). A more thorough breakdown is essential to fully understand the role and necessity of each part.

3. The performance gains of Sherlock over strong baselines such as LLaVA-CoT and LlamaV-o1 are incremental, especially in the earlier training stages. Much of the observed improvement arises from the iterative self-improvement phase, which the baselines do not employ. This makes it unclear whether the gains stem from the framework itself or simply from the additional training procedure.

---

> ### Author Rebuttal · Authors · 2025-07-31
>
> Thanks for your valuable feedback, which greatly contributed to improving the paper!
>
> > Weakness 1: A major concern is the limited discussion of related work. While several components of Sherlock's algorithm appear to draw on techniques recently explored in the LLM literature, the paper lacks a clear delineation of which aspects are novel or uniquely adapted to the multimodal setting. It would greatly benefit from explicitly clarifying what parts of the method are specific to vision-language reasoning.
>
> - **Novel design of Sherlock.** Sherlock introduces several novel designs, differentiating itself from related works such as ARIES [1] and "Self-rewarding correction for mathematical reasoning" [2]. Key distinctions are summarized below:
>
>     1.  **Domain Difference:** While [1] and [2] are designed for LLMs, Sherlock is specifically tailored for VLMs.
>
>     2.  **The First Trajectory-Level Self-Correction Objective.** Previous works like [2] construct preference datasets based solely on final answer verification (i.e., chosen if the final answer is correct, rejected otherwise). [1] relies on an external reward model to judge the best and worst responses. We argue that for reasoning models, such coarse-grained preference signals are often insufficient. For example, some initial reasoning steps within a rejected response might be superior to those in a chosen response, even if the final answer is incorrect. To address this, we propose a more fine-grained **trajectory-level learning objective**. Instead of requiring a complete rewrite of the response, our model is guided to revise only the erroneous suffix. This approach prevents training from being perturbed by prefixes lacking clear preference signals. Experiments in Figure 4 and analysis in Finding 2 demonstrate that our trajectory-level objective yields superior self-correction capabilities in reasoning VLMs.
>
>
>     3.  **Novel Strategy for Preference Data Generation.** Unlike [1] and [2], which rely on rule-based verification or reward models to construct preference datasets, Sherlock fully leverages the introduced vision modality. Our approach involves randomly truncating the *i*-th reasoning step in a chosen response, then injecting random noise into the image. We then prompt the VLM to generate based on the question, the perturbed image, and the prefix reasoning steps. This method easily yields low-quality suffix reasoning that often contains visual hallucinations or misunderstandings. Crucially, our data construction pipeline requires no external verification or reward signals, a limitation that restricted [1] to easily verifiable math tasks and [2] to alignment tasks.
>
>
>     4.  **Introduction of Sample-Level Dynamic $\beta$ for Stable Preference Training.** Inspired by $\beta$-DPO [3], we extend their batch-level dynamic $\beta$ to a **sample-level dynamic $\beta$** to accommodate the varying quality gaps inherent in our preference data construction. We precisely control the quality gap for each sample by randomly truncating the reasoning step $i$ and injecting random noise $\epsilon$ into the image. This innovative use of $\epsilon$ and $i$ to derive a sample-specific dynamic $\beta$, detailed in Equation 8, significantly stabilizes preference training, as evidenced by experiments in Figure 4 and analysis in Finding 3.
>
>     5.  **Key Takeaways and Interesting Findings.** In Section 2, we are the first to highlight that reasoning VLMs cannot perform effective self-correction, whether step-wise or response-wise. Our experimental findings further reveal that reasoning and self-correction are not orthogonal capabilities; learning one positively influences the learning process of the other.
>
>     6.  **Novel Scaling Strategy for Self-Correction Inference.** In Section 5.3, we propose a novel strategy to scale up self-correction inference by introducing a verification model to control the correction process. Correction continues until the verification model deems the response correct. If the judgment is incorrect, the corresponding critique is incorporated into the self-correction prompt as an external signal. As demonstrated by experiments in Table 4, this integrated verification approach significantly accelerates inference-time self-correction reasoning (reducing GPU hours from 13.2 to 8.7) and improves accuracy (from 54.0 to 55.9). This combination also substantially improves upon the traditional use of verification models in parallel majority vote settings, increasing accuracy from 55.1 to 55.9 and drastically reducing inference GPU hours from 40.2h to 8.7h.
>
> - **Aspects are uniquely adapted to the multimodal setting.** Based on our previous analysis, points 3 and 4 are uniquely adapted to the multimodal setting, which makes Sherlock more suitable in a multimodal setting.
>
> > Weakness 2: Sherlock's multi-stage training pipeline—comprising SFT, offline preference training, and iterative self-improvement—adds significant complexity. However, the paper provides limited discussion on how this design might be deployed or tuned in practice. Moreover, while an ablation study is provided, it does not clearly isolate the contribution of each individual loss component (e.g., DPO vs. self-correction). A more thorough breakdown is essential to fully understand the role and necessity of each part.
>
> 1. **Sherlock is not complex during training.** Sherlock's training process is more efficient than that of the baseline models. Our methodology utilizes a streamlined pipeline consisting only of SFT and non-RL preference training, which enhances reasoning and self-correction capabilities without the need for more complex stages like multi-turn conversation training used by methods such as LlamaV-o1. To provide concrete evidence, the table below reports the total A100 GPU hours required for the entire training process of Sherlock and the baselines. All training was conducted on an 8xA100 GPU node using DeepSpeed ZeRO-3. The results clearly indicate that Sherlock's overall training time is substantially lower, highlighting its computational efficiency.
>
>     |Methods|A100 GPU Hours|
>     |-|-|
>     |LLaVA-CoT|~160|
>     |LlamaV-o1|~288|
>     |Sherlock|~128|
>
> 2. **Implementation and Reproducibility.** To ensure Sherlock can be easily deployed in practice, **we have provided comprehensive implementation details in the paper**. The loss functions are detailed in Equations 3, 6, and 7. We also outline specific training details, including the device, framework, and all hyperparameters, in Section 5 and Appendix B.2 (Table 5). Moreover, a detailed algorithm for Sherlock's training pipeline is presented in Appendix B.2 on page 18 to facilitate reproducibility.
>
> 3. **We clearly isolate the contribution of each individual loss component (e.g., DPO vs. self-correction) in an ablation study and provide corresponding analysis.** We conducted a thorough ablation study to clearly isolate the contribution of each individual loss component. As shown in Figure 4, we present results for models trained without the DPO loss (term w/o $\mathcal L_{\text{DPO}}$) and without the self-correction loss (term w/o $\mathcal L_{\text{SC}}$). Our analysis in "Finding 1" reveals a key insight: the capabilities learned from self-correction ($\mathcal L_{\text{SC}}$) and reasoning ($\mathcal L_{\text{DPO}}$) are not orthogonal. Instead, they appear to be intertwined and mutually beneficial during the training process.
>
> > Weakness 3: The performance gains of Sherlock over strong baselines such as LLaVA-CoT and LlamaV-o1 are incremental, especially in the earlier training stages. Much of the observed improvement arises from the iterative self-improvement phase, which the baselines do not employ. This makes it unclear whether the gains stem from the framework itself or simply from the additional training procedure.
>
> It is worth noting that Sherlock achieves better performance with remarkable data efficiency. We randomly sampled just 20k annotated data points and 10k unannotated questions from the LLaVA-CoT dataset. Using only these 20k samples for the SFT and offline preference training stages, Sherlock achieves performance comparable to the original LLaVA-CoT across eight different benchmarks, even though LLaVA-CoT was trained on the full 100k annotated samples. Moreover, our offline model can perform stable self-correction to improve its reasoning—a capability that LLaVA-CoT struggles with. To ensure a fair comparison, we retrained LLaVA-CoT using the exact same 20k data subset. As shown in the table below, Sherlock significantly outperforms LLaVA-CoT under these identical data conditions and successfully unlocks the self-correction ability of the base VLM, demonstrating the superiority of our training pipeline. Furthermore, the incremental performance gains observed during online iterative training further validate that Sherlock can self-enhance its reasoning and self-correction abilities without requiring any external guidance or annotated data.
>
> |Methods|# Data w/ GT|MMB|MMVet|Hallus|MMMU|MMStar|AI2D|MathV|MME|Avg.|
> |-|-|-|-|-|-|-|-|-|-|-|
> |LLaVA-CoT|100k|75.0|61.7|47.7|49.1|57.6|82.9|53.7|2177|63.2|
> |+ Self-Correction|100k|74.4|62.3|46.4|49.2|57.8|82.9|53.0|2183|63.0$^{↓0.2}$|
> |LLaVA-CoT-20k|20k|72.0|58.9|47.9|47.9|55.3|88.3|48.7|2158|62.0|
> |+ Self-Correction|20k|71.6|58.6|46.8|48.7|56.0|87.7|49.8|2109|61.8$^{↓0.2}$|
> |Sherlock Offline|20k|73.2|61.4|48.1|47.6|57.5|88.4|52.2|2162|63.2|
> |+ Self-Correction|20k|74.7|63.8|48.9|49.0|57.7|89.5|53.9|2171|64.4$^{↑1.2}$|
>
> [1] Zeng et al; ARIES: Stimulating Self-Refinement of Large Language Models by Iterative Preference Optimization
>
> [2] Xiong et al; Self-rewarding correction for mathematical reasoning.
>
> [3] Wu et al; $\beta$-DPO: Direct Preference Optimization with Dynamic $\beta$

---

> > ### Comment · Reviewer_sFen · 2025-08-08
> >
> > The response has addressed most of my previous concerns and questions. However, as I am not very familiar with this area, I prefer to keep my original score.

---

> > > ### Author Response · Authors · 2025-08-09
> > >
> > > Thank you for the response. We truly appreciate your valuable insights, which have helped us improve the quality of our work. We will incorporate the suggested analysis in our revision. Thank you for your support of our paper.

---

### Official Review · Reviewer_72iE · 2025-06-24

**Clarity:** 2
**Significance:** 2
**Originality:** 2
**Rating:** 4
**Confidence:** 2

**Summary:**

This paper addresses the challenges of reasoning Vision-Language Models (VLMs), such as sensitivity to reasoning errors, high data dependency, and poor generalization, by introducing Sherlock, a self-correction and self-improvement training framework. Key innovations include a trajectory-level self-correction objective (correcting only erroneous steps), a visual perturbation-based method to construct preference datasets with controllable quality gaps, and a dynamic β for stabilizing preference training. Sherlock requires only 20k randomly annotated data for initialization, then improves unsupervised via online iterative self-generated preference pairs. Experiments show that Sherlock Iter2 on Llama3.2-Vision-11B achieves 64.1% direct reasoning accuracy and 65.4% after self-correction across 8 benchmarks, outperforming LLaVA-CoT, Mulberry, etc., with less than 20% of their data. Combining with a verifier reduces GPU usage by 40% while enhancing accuracy.

**Questions:**

1.How to override underlying representation errors when reasoning errors originate from visual misinterpretation?
2.Can dynamic β adapt to quality gaps of different error types? Are there optimal adaptive mechanisms?
3.How to define convergence criteria for online iteration to avoid overfitting to self-generated errors?
4.Why does trajectory-level correction fail in cross-modal reasoning? How to extend to temporal/3D modalities?
5.How does verifier misjudgment affect correction directions? How to build a robust verification-correction loop?

**Ethical Concerns:**

["NO or VERY MINOR ethics concerns only"]

**Final Justification:**

Thanks for your detailed response, which has resolved my concerns. After also carefully reviewing the comments from the other reviewers, I prefer to keep my original score.

**Limitations:**

yes

**Paper Formatting Concerns:**

None.

**Quality:**

2

**Strengths And Weaknesses:**

Advantages​
1.The research directly addresses the core challenges of reasoning Visual Language Models (VLMs), including error propagation, data dependency, and insufficient generalization capabilities. Through a meticulously designed self-correction mechanism, it effectively reduces errors generated during multi-step reasoning processes, and this achievement has significant practical implications in real-world application scenarios.​

2.The trajectory-level correction strategy takes an unconventional approach by only amending the erroneous suffix part instead of completely discarding the entire response. This method aligns with the characteristics of auto-regressive decoding, allowing it to retain the correct steps in the reasoning process while significantly reducing computational costs. Additionally, the use of visual perturbation in constructing preference data makes full use of multimodal features to artificially create controllable quality differences, opening up a new path for unsupervised self-improvement.​

Disadvantages​
1.The effectiveness of trajectory-level correction depends on the accurate localization of the erroneous suffix. However, when the root cause of the error lies in a fundamental misunderstanding of visual content, such as semantic errors occurring during image understanding, due to incorrect initial visual feature extraction, this correction method may not function properly.​

2.The dynamic β parameter was originally designed to address quality differences caused by random visual perturbations. However, when facing the complex and diverse types of errors in the real world, such as semantic errors, logical errors, and calculation errors, it is difficult to flexibly adapt. This may lead to situations of under-correction or over-correction, affecting the accuracy and stability of the model.​

3.The online iteration process lacks clear convergence criteria, such as when the accuracy rate stabilizes or when the generation diversity reaches a threshold. This exposes the model to the risk of overfitting to its own generated error patterns, and currently, there is no theoretical guarantee of stability, making it difficult to ensure the reliability of the model during the iteration process.

---

> ### Author Rebuttal · Authors · 2025-07-31
>
> We sincerely thank the reviewer for the thoughtful and thorough comments on our paper. We believe your feedback will improve the paper.
>
> > Weakness 1 & Question 1: The effectiveness of trajectory-level correction depends on the accurate localization of the erroneous suffix. However, when the root cause of the error lies in a fundamental misunderstanding of visual content, such as semantic errors occurring during image understanding, due to incorrect initial visual feature extraction, this correction method may not function properly.​
>
> 1.  **Trajectory-Level Correction Doesn't Require Localizing Erroneous Suffixes.** Our training objective focuses on trajectory-level correction, which trains the model to revise only the problematic suffix of a generated response. During training, we don't need to explicitly localize this suffix. We achieve this by creating a preference dataset using controlled, random *i*-step truncation and injecting random visual noise. This process generates a low-quality reasoning suffix ($Y_{\geq i}^l$).
>
> 2.  **Sherlock Uses a Fixed Prompt for Internal Self-Correction.** During inference, Sherlock first generates an initial direct reasoning response. It then concatenates this initial response with the original question to form a specific **correction prompt** (detailed in Appendix C.4). For correction, Sherlock does not need to pinpoint error steps in the previous response; it directly generates a new reasoning response based on this correction prompt. This design means Sherlock's correction behavior at inference time is **inherent and doesn't require localizing** the erroneous suffix.
>
> 3.  **Sherlock Effectively Corrects Visual and Knowledge Errors.** Our training methodology involves adding random noise to input images. This produces low-quality, rejected reasoning suffixes that contain visual misunderstandings and hallucinations. Therefore, our preference training **explicitly teaches the model to correct errors related to image understanding**. Appendix E provides case studies illustrating this self-correction. For example, in Case 3's direct generation, the `<CAPTION>` stage demonstrated a fundamental visual content misunderstanding: "It appears to have been cut into several slices, with two main cuts extending from the center towards the edge, suggesting it is divided into halves." However, Sherlock successfully corrected this error in subsequent self-correction turns using its internal self-correction prompt and obtained the final right answer. Case 4 further shows Sherlock's ability to correct both visual recognition and knowledge errors during self-correction.
>
> ***
>
> > Weakness 2 & Question 2: The dynamic $\beta$ parameter was originally designed to address quality differences caused by random visual perturbations. However, when facing the complex and diverse types of errors in the real world, such as semantic errors, logical errors, and calculation errors, it is difficult to flexibly adapt. This may lead to situations of under-correction or over-correction, affecting the accuracy and stability of the model.​
>
> 1.  **Dynamic $\beta$ Stabilizes Sherlock's Preference Training.** The dynamic $\beta$ parameter is designed to adapt to the varying quality gaps between samples in our preference dataset, which is specific to the Sherlock pipeline. We generate this dataset with controllable sample-level quality gaps by randomly sampling the truncation step $i$ and injecting random noise $\epsilon$ into the images. This allows us to design a sample-specific dynamic $\beta$ based on $i$ and $\epsilon$, providing a "free lunch" to control optimization updates and leading to more stable preference training. Our ablation study in Figure 4 compares dynamic $\beta$ with a vanilla static $\beta$, demonstrating that dynamic $\beta$ yields superior reasoning and self-correction performance.
>
> 2.  **Error type is not related to design of $\beta$.** We believe the types of errors a model can correct are not related to the design of $\beta$. Instead, they are more closely related to the pre-trained abilities of the base VLM and the specific data types used during Sherlock's training. As illustrated in Figure 1, Sherlock can also correct logical errors (marked in red), leading to the correct answer.
>
> ***
>
> > Weakness 3 & Question 3: The online iteration process lacks clear convergence criteria, such as when the accuracy rate stabilizes or when the generation diversity reaches a threshold. This exposes the model to the risk of overfitting to its own generated error patterns, and currently, there is no theoretical guarantee of stability, making it difficult to ensure the reliability of the model during the iteration process.
>
>
> This is an open question in the whole field of LLM self-improvement or self-training. All methods that involve training on self-generated data are subject to the possibility you mentioned, so this is not a problem unique to Sherlock. In Sherlock, we conduct self-improvement training motivated by ARIES [1] in the LLM domain, and the natural quality gap between and after self-correction, which makes it easy to construct a preference pair without an external signal. Therefore, we follow the online training iteration settings in AIRES and other self-improvement works in the LLM domain, setting the online iteration to 2. Results presented in Table 2 demonstrate that Sherlock consistently improves its reasoning and self-correction abilities using its self-constructed preference dataset. This empirically validates that Sherlock effectively avoids overfitting to self-generated data across two online iterations.
>
> > Question 4: Why does trajectory-level correction fail in cross-modal reasoning? How to extend to temporal/3D modalities?
>
> We do **not** claim that trajectory-level correction fails in cross-modal reasoning. Intuitively, Sherlock can be extended to any input modalities, if the model performs autoregressive reasoning based on text. The only necessary modification would be to inject different types of noise during preference data construction. For instance, one could apply Gaussian noise to images (both RGB and depth), white noise to audio, and similar perturbations to other data types. Exploring different modalities is beyond the scope of this work and is left for future research.
>
> > Question 5: How does verifier misjudgment affect correction directions? How to build a robust verification-correction loop?
>
> Sherlock operates on a fully internal self-correction mechanism, **without the need for a separate verification step before correction**. During inference, Sherlock initially generates a direct reasoning response. This response is then concatenated with the original question to form a specific **correction prompt** (detailed in Appendix C.4). For the correction phase, Sherlock does not need to identify specific error steps in its previous response; instead, it directly generates a new reasoning response guided by this comprehensive prompt. Consequently, our approach does not need a distinct verification-correction loop.
>
> Regarding the verification model discussed in Section 5.3, we view it as a promising direction to both scale and accelerate self-correction generation. While vanilla self-correction in our experiments is fixed at three turns, incorporating a verification model allows us to dynamically control the self-correction process. Correction continues until the verification model deems the response correct. If the judgment is incorrect, the corresponding critique is incorporated into the self-correction prompt as an external signal. As demonstrated by experiments in Table 4, this integrated verification approach significantly accelerates inference-time self-correction reasoning (reducing GPU hours from 13.2 to 8.7) and improves accuracy (from 54.0 to 55.9). Therefore, we believe this presents a promising avenue for further scaling self-correction reasoning and is a valuable direction for future work.
>
> [1] Zeng et al; ARIES: Stimulating Self-Refinement of Large Language Models by Iterative Preference Optimization

---

> > ### Comment · Area_Chair_cDkk · 2025-08-06
> >
> > Dear Reviewer 72iE,
> >
> > This is a reminder that we are nearing the end of the author-reviewer discussion period. Please carefully read the authors' responses to your reviews, as well as the responses to other reviews. If you have any follow-up questions, please post them ASAP, so there is time for back-and-forth discussion with the authors.
> >
> > If you don’t have any additional questions and if your concerns have been addressed, please update your ratings or final justification accordingly. If you are not willing to update your ratings, please explain in your final justification why concerns are still not addressed.
> >
> > Note that the author-review discussion period will end on **Aug 8, 11.59pm AoE**.
> >
> > Best,
> >
> > AC

---

> > ### Comment · Reviewer_72iE · 2025-08-06
> >
> > Thanks for your detailed response, which has resolved my concerns. After also carefully reviewing the comments from the other reviewers, I will maintain my borderline-accept score.

---

> > > ### Author Response · Authors · 2025-08-08
> > >
> > > Thank you for your response. We are glad our rebuttal addressed your concerns. We would appreciate it if you could consider raising your score to reflect this. Please let us know if you have any further questions. Thank you for your thoughtful feedback.

---

### Official Review · Reviewer_aRRH · 2025-06-30

**Clarity:** 3
**Significance:** 2
**Originality:** 2
**Rating:** 4
**Confidence:** 4

**Summary:**

This paper introduces Sherlock, a model designed to enhance VLMs with self-correction capabilities for multimodal reasoning. The authors begin with a preliminary study highlighting the limitations of current reasoning VLMs (e.g., LLaVA-CoT and VL-Rethinker), showing that these models struggle with both step-wise and response-level self-correction. To address this, Sherlock focuses on enhancing response-level self-correction through a multi-stage training framework: (1) a supervised fine-tuning (SFT) cold-start phase to strengthen response-level correction; (2) offline preference learning that targets erroneous reasoning steps for trajectory-level correction; and (3) iterative online preference training, which leverages both original and corrected responses from the previous Sherlock model to further improve self-correction. Experiments using LLaMA3.2-Vision-11B as the base model demonstrate that Sherlock outperforms existing baselines on several visual reasoning benchmarks.

**Questions:**

Besides the questions I raised in weaknesses part:

- The trajectory-level objectives proposed in Sherlock is similar to recent step-wise DPO methods. It would be beneficial to more thoroughly discuss the connection and differences between Sherlock and these step-wise DPO approaches.
- The right side of Figure 2 is difficult to interpret at first glance. It feels unintuitive to present results from different models (e.g., LLaVA-CoT and Qwen) in the same plot without clear separation or explanation. Consider revising the figure to improve clarity.

**Ethical Concerns:**

["NO or VERY MINOR ethics concerns only"]

**Final Justification:**

The response has addressed most of my previous concerns and questions. I will raise my score to a 4.0.

**Limitations:**

The limitations discussed in Section E are relatively brief and lack depth. Consider expanding this section to provide a more thorough reflection on the method’s constraints and potential areas for future improvement.

**Quality:**

3

**Strengths And Weaknesses:**

## Strengthes

- The paper addresses a timely and important problem—enhancing reasoning and self-correction capabilities in VLMs. The task is well-motivated, and the preliminary study provides insightful observations, showing that current VLMs (e.g., LLaVA-CoT and VL-Rethinker) struggle with both step-wise and response-level self-correction.
- Building upon these findings, the authors propose a three-stage training framework that progressively improves self-correction. The approach is plausible, tackles both response- and trajectory-level correction, and remains relatively data-efficient, requiring only 20K samples.
- The experimental results and analyses demonstrate clear improvements over baselines, and the ablation studies are comprehensive, effectively validating the contribution of each component.

## Weaknesses

- While the preliminary analysis is insightful, it relies on relatively small models (LLaVA-CoT and VL-Rethinker), which may limit the generalizability of the findings.
- Sherlock primarily focuses on response-level self-correction, which shares similarities with self-refinement methods, where a model revises an initial response to improve accuracy. Thus, it would strengthen the work to include comparisons with self-refinement baselines—such as prompting a larger VLM (e.g., GPT) for multi-round refinement—or sampling-based methods like Best-of-N generation and reranking.
- The analysis could benefit from a deeper exploration of error types—e.g., which types of reasoning errors Sherlock can or cannot correct, and how performance evolves across multiple rounds of self-correction. Such insights would offer greater value to the community.
- Several technical details remain unclear: (1) In Line 130, how is the "Modify One Step" operation implemented? Is Qwen explicitly prompted to alter a correct step into an incorrect one? (2) In Table 2, does Sherlock generate outputs in a two-stage process (generation followed by self-correction)? If so, are the Sherlock variants in the Table 2 results before self-correction, and the "+ Self-correction" versions the revised results? (3) Related to the above, in Table 3, are the critic models used for self-correction response-level or step-wise? The fact that Sherlock + Self-correction underperforms Qwen2.5-VL-7B on some benchmarks suggests that error detection and correction remain challenging, even with training (while Qwen2.5-VL-7B is not trained). (4) Regarding model choice, why is MLLAMA used as the base model instead of Qwen-VL?

---

> ### Author Rebuttal · Authors · 2025-07-31
>
> Thanks for your valuable feedback, which greatly contributed to improving the paper!
> > W1
>
> Before our paper submission, most open-source reasoning VLMs typically ranged from 3B to 11B parameters, and our initial analysis primarily focused on this popular model size range. To extend our findings, we conducted the same experiments, "Modify One Step", "Qwen2.5-VL-7B Critic", and "Self-Correction", on the larger VL-Rethinker-72B model.
>
> The results, presented in the following tables on the MMStar benchmark, align with our earlier findings from Sec. 3. Even this substantially larger 72B reasoning model fails to effectively self-correct its reasoning responses, whether in a step-wise or response-wise manner. This demonstrates the generalizability of our key takeaways, indicating that the inherent limitations in self-correction extend beyond smaller VLM architectures.
>
> |Method|MMStar|
> |-|-|
> |VL-Rethinker-72B|71.3|
> |+ Modified One Step|33.7|
> |+ Qwen2.5-VL-7B-Instruct|70.9|
> |+ Self-Correction|70.0|
>
> ***
>
> > W2
>
> As we observe and report in the right panel of Fig. 2, reasoning models such as LLaVA−CoT struggle with self-refinement, even when aided by external critics like Critic-V and Qwen2.5-VL-Instruct-7B. To further investigate this, we conducted a more rigorous evaluation. We employed the more powerful Qwen2.5-VL-Instruct-32B model to provide external critiques for LLaVA-CoT's responses. This process was extended over **three correction turns**, with Qwen2.5-VL-Instruct-32B offering feedback in each turn to guide LLaVA-CoT's revisions. We performed these experiments across eight benchmarks to thoroughly assess the impact of these external signals on LLaVA-CoT's performance. Results further demonstrate that LLaVA-CoT struggles to improve its reasoning performance through self-refinement, even when critiques are provided by a large and powerful 32B VLM. In contrast, Sherlock consistently achieves stable self-correction without requiring any external signals.
>
> |Model|MMB|MMVet|Hallus|MMMU|MMStar|AI2D|MathV|MME|Avg.|
> |-|-|-|-|-|-|-|-|-|-|
> |LLaVA-CoT|75.0|61.7|47.7|49.1|57.6|82.9|53.7|2177|63.2|
> |+Qwen2.5-VL-32B 3 Turns Critic|76.3|61.8|44.5|50.0|58.0|81.4|54.0|2200|63.1|
> |+Self-Correction|74.4|62.3|46.4|49.2|57.8|82.9|53.0|2183|63.0|
> |Sherlock|74.6|62.4|48.7|49.7|57.7|89.6|52.0|2197|64.1|
> |+Self-Correction|77.2|62.6|51.2|50.1|59.0|90.6|54.0|2204|65.4|
> ***
> > W3
> 1. **Reasoning Errors Analysis.** As demonstrated in our case studies (Fig. 1 and Appendix E), Sherlock is particularly effective at correcting false reasoning that stems from either visual hallucinations or logical mistakes.
> - **Visual Hallucination Correction.** Its proficiency in correcting visual hallucinations is a direct result of our preference data construction. By intentionally introducing visual errors into the negative responses, our training explicitly teaches VLM to identify and self-correct these specific types of error. This capability is evident in the examples provided, such as correcting errors in color perception (Case 1), object counting (Case 3), and person recognition (Case 4).
> - **Logic Mistake Correction.** We believe Sherlock's success is due to an inherent advantage of self-correction. Intuitively, it is a simpler task for the model to recognize a conspicuous logical flaw in its own prior response and correct it than it is to generate a perfectly coherent reasoning chain from scratch.
>     While generally robust, Sherlock still exhibits failure cases with difficult samples, including some complex mathematical problems. We will add some case studies in the revision. As Sherlock randomly sampled a small amount of data from LLaVA-CoT dataset, we posit that the domain of the training data impacts Sherlock's capabilities.
> 2. **Performance Evolving across multiple rounds of self-correction.** We have evaluated Sherlock's performance over three rounds of self-correction. The detailed performance evolution curve, presented in Appendix D, Fig. 5, clearly demonstrates Sherlock's ability to achieve stable and effective inference-time scaling through iterative self-correction.
> ***
> > W4
> 1.  **Modify One Step Setting.** Yes, the “Modify One Step” operation is implemented by explicitly prompting Qwen2.5 to transform one intermediate reasoning step into an incorrect version. Specifically, we first generate the initial reasoning response $ \\{y_1, \cdots, y_n;a\\} \sim \pi (x_{I\\&T}) $. Then, we prompt Qwen2.5 to transform one of the intermediate steps $y*\in\{y_1,\cdots,y_{i-1}\}$ into an incorrect step $\hat{y*}$ (Qwen autonomously selects which step to modify; detailed in Appendix C.3). Finally, we condition the reasoning VLM on the new, modified prefix $ \\{ y_1,\cdots,\hat{y*},\cdots,y_{i-1} \\} $. The model then autoregressively generates a new suffix and final answer, yielding $\\{\hat{y_i}\cdots,\hat{y_n};\hat{a}\\}\sim\pi(x_{I\\&T}; y_1,\cdots,\hat{y*},\cdots,y_{i-1})$. We set the truncation step, denoted as i, to the middle step for each reasoning response. This prevents Qwen2.5 from directly modifying later reasoning steps that might contain or be highly correlated with the final answer.
> 2.  **Sherlock Generation Process.** Yes, Sherlock first generates the direct generation response, which corresponds to the performance of Sherlock Variants in Table 2. Then, to achieve "+ Self-Correction" results in Table 2, we concatenated the original question with the previously generated response to create a correction prompt (detailed in line 493 of Appendix B.1), which forced Sherlock to perform response-wise corrections over three turns.
> 3.  **Critic Model in Table 3.** In Table 3, all critic models provide a critique for the *entire* response. We then concatenate the corresponding critique with the previously generated responses into a correction prompt (detailed in Appendix C.4); thus, this should be considered response-wise correction. When using Qwen2.5-VL-7B as the critic, Sherlock’s performance after correction exceeds its self-correction baseline on some benchmarks. We consider this an advantage of our training framework, as it enhances the self-correction ability of VLMs, making them more sensitive to external critique. This implies that a high-quality critique can significantly boost Sherlock's performance. Conversely, as analyzed and observed in Sec. 3, using Qwen2.5-VL as a critic model does not improve LLaVA-CoT's performance, as most VLMs are not inherently responsive to external feedback. We see Sherlock’s ability to benefit from additional critics as an advantage, not a limitation, enabling flexible inference-time scaling and further performance gains.
> 4.  **Model Choice.** Sherlock is developed based on the LLaVA-CoT dataset, which serves as a SFT baseline. Given that Sherlock's training primarily involves SFT and preference training, and does not rely on RL, we opted not to conduct experiments with Qwen-VL, a model typically employed for RL-based reasoning training [1][2]. Instead, to ensure a fair comparison and validate Sherlock's superior reasoning and self-correction performance, we compared it against multiple SFT methods (LLaVA-CoT, LlamaV-o1, and Mulberry), all of which were initialized from the same Llama3.2-Vision base model. We plan to consider incorporating Qwen-VL in future work.
> ***
> > Q1
>
> Our trajectory-level objective differs from step-DPO [3] in two key ways, which we highlight below:
> 1.  **Learning objective of Step-DPO is only a small part of Sherlock.** Step-DPO primarily provides step-level preference datasets where chosen and rejected responses share an identical prefix, but the chosen response's suffix quality surpasses that of the rejected one. Its optimization objective is the same as vanilla DPO: maximizing the likelihood of the chosen response while minimizing that of the rejected one. In contrast, our objective considers both direct generation and self-correction, which involves more learning directions than direct generation in vanilla DPO. For example, the self-correction objective in our approach:
>     * Discourages the failure to correct a low-quality response (i.e., outputting the same suboptimal response).
>     * Discourages "mis-correcting" a high-quality response into a lower-quality one.
>     * Encourages the correction of a low-quality response into a high-quality one.
>     * Encourages the retention of an already high-quality response.
>
>     Given that our chosen and rejected responses often have different prefixes, our framework encompasses a total of eight learning directions, which provides more learning signals than Step-DPO.
> 2.  **Differing Prefix Steps.** A crucial difference lies in the prefixes of chosen and rejected responses. In Sherlock online iteration, these prefixes are distinct, whereas step-DPO's prefix ($s_{1\sim k-1}$) remains constant. While a shared prefix does not impact DPO's loss computation—making step-DPO's optimization identical to DPO's despite its step-wise datasets—Sherlock's approach is different. Our prefixes are not identical but possess comparable quality, as evidenced by our experiments and analysis in Sec. 5.2, Finding 2. This design for the trajectory loss term specifically only modifies the low-quality suffix $Y_{\geq i}^l$ identified during data construction, thereby avoiding updates on prefix steps $Y_{<i}^l$ and $Y_{<i}^w$, which lack clear preference signals.
> ***
> > Q2
>
> We have annotated the “LLaVA-CoT Acc” and “VL-Rethinker Acc” directly within the corresponding sub-figure on the right side of Fig. 2 in the submitted version. We will further enhance the clarity of this figure in the revised manuscript.
> ***
> > Limitations.
>
> Thank you for the valuable suggestion. We will extend our analysis and discussion to enhance the quality and depth of the Limitations in our revised manuscript.
>
> [1] Wang et al; VL-Rethinker
>
> [2] Chen et al; SFT or RL? An Early Investigation into Training R1-Like Reasoning Large Vision-Language Models
>
> [3] Lai et al; Step-DPO

---

> > ### Comment · Reviewer_aRRH · 2025-08-05
> >
> > Thanks for your detailed response, which has addressed most of my previous concerns and questions. I will raise my score to a 4.0.

---

> > > ### Author Response · Authors · 2025-08-05
> > >
> > > Thank you very much for your positive feedback. Your insightful comments have greatly improved our work. We will revise our manuscript based on your suggestion. We sincerely appreciate your support!

---

### Official Review · Reviewer_Kt1S · 2025-07-02

**Clarity:** 3
**Significance:** 2
**Originality:** 2
**Rating:** 4
**Confidence:** 4

**Summary:**

This paper presents a framework, called Sherlock, for enhancing the reasoning ability of vision-language models (VLMs) through self-correction and iterative self-improvement. The proposed method focuses on fine-grained correction of reasoning trajectories, using preference optimization techniques to teach VLMs to revise only incorrect steps instead of rewriting full responses. The work is evaluated across eight benchmarks and demonstrates strong performance with significantly less annotated data.

**Questions:**

1. Could you please answer to my questions/concerns in the first and second item listed on the Weaknesses panel?

2. Could the author(s) provide the experimental results of which are initialized from *-CoT models in the first stage?

**Ethical Concerns:**

["NO or VERY MINOR ethics concerns only"]

**Final Justification:**

The authors' response has addressed most of my concerns regarding this work. While the lack of scaling analysis for the selected training dataset means we still miss a thorough investigation of the method's performance ceiling, the overall improvements and clarifications are clear. Overall, I have revised my score from 2 to 4.

**Limitations:**

yes

**Paper Formatting Concerns:**

The absence of a dedicated Conclusion section in the main text is a structural flaw (just briefly discussed in the appendix).

**Quality:**

2

**Strengths And Weaknesses:**

# Strengths
- Clear motivation: The paper identifies a meaningful limitation of existing reasoning VLMs—namely, their inability to recover from intermediate errors. This motivates the proposal of self-correction as a mechanism to improve robustness and generalization.

- Empirical efficiency: Despite its modest performance gains, Sherlock achieves them using significantly fewer annotations (20k) than baselines like LLaVA-CoT or Mulberry, which supports the claim of improved annotation efficiency.


# Weaknesses
### Cold-Start Strategy is not Thoroughly Justified
In Stage I, the authors fine-tune a base VLM on 10k samples randomly selected from the LLaVA-CoT dataset to initiate the Sherlock pipeline. However, this decision appears suboptimal and insufficiently justified. There exist multiple CoT-pretrained VLMs, such as LLaVA-CoT and LlamaV-o1, which are trained on much more extensive data. Using one of these as the starting point would likely result in a better initialization for downstream correction learning. The rationale for not leveraging these stronger checkpoints as a base model remains unclear and undermines the significance of Sherlock's first-stage improvements.

### Unvalidated Noise Injection in Data Construction
The second stage constructs negative preference samples by injecting random visual noise and truncating reasoning trajectories. However, the paper lacks any quantitative analysis or diagnostics regarding how similar or dissimilar the positive and negative examples are. If the noise degrades the negative samples too aggressively, the quality gap may become too large to provide meaningful learning signals, essentially turning the preference learning into trivial classification. This weakens the effectiveness of the training objective, especially when fine-grained trajectory-level reasoning is the goal.

### Limited Comparison with Recent and Relevant Work
The empirical comparisons do not sufficiently cover recent state-of-the-art approaches in multimodal reasoning, particularly those involving self-training via reflection, self-train via reasoning, structured prompting, etc. This omission raises concerns about the strength and generalizability of Sherlock’s reported improvements. List some here but not limited:
- Vision-Language Models Can Self-Improve Reasoning via Reflection
- Conme: Rethinking evaluation of compositional reasoning for modern vlms
- Improve Vision Language Model Chain-of-thought Reasoning

### Modest Performance Gain over Strong CoT Baselines
From the main results in Table 2, the performance improvements over strong CoT-based models (e.g., LlamaV-o1) are relatively modest (e.g., 63.4 --> 65.4 average accuracy), despite the added training complexity. This raises an important question: If Sherlock had started from a stronger base model such as LlamaV-o1, would the additional self-correction stages still yield measurable improvements? This missing analysis leaves open the possibility that Sherlock's contribution may simply stem from better pretraining or CoT exposure, rather than its self-correction mechanism.

### Minor Weakness
- Ethical considerations of using self-generated preference data (e.g., bias propagation, hallucinated corrections) are not discussed.
- The absence of a dedicated Conclusion section in the main text is a structural flaw (just briefly discussed in the appendix).

---

> ### Author Rebuttal · Authors · 2025-07-31
>
> Thank you for your great efforts on the review and constructive comments.
> > W1: Cold-Start Strategy
> - **Goal of Sherlock.** The goal of Sherlock is **not merely to improve upon existing CoT models**, but to introduce a new framework that transforms a base VLM into a reasoning VLM with both strong reasoning and self-correction capabilities. Initializing from a CoT model would conflate this objective. Our analysis reveals that existing reasoning VLMs struggle to self-correct their results, even when trained on large-scale datasets that include reflection steps. Sherlock aims to use the same data to jointly enable reasoning and self-correction, offering a new paradigm for training reasoning VLMs that is in parallel to, rather than building upon, existing CoT training methods.
> - **Problems with Initialization on LLaVA-CoT.** Fine-tuning LLaVA-CoT on a subset of their original training data risks **severe overfitting** because the model has already extensively seen these samples. This leads to two issues:
>     1. The model will confidently generate the already-memorized answer, offering a negligible learning signal for the self-correction objective in Eq. 3. With high certainty in its initial prediction, the model has little room to learn reasoning refinement.
>     2. The direct generation objective suffers from overfitting, as LLaVA-CoT, heavily trained on $(x,Y^w)$, limits its generalization to unseen data.
>
>     While a different dataset could help, constructing one is expensive. Our empirical validation, fine-tuning LLaVA-CoT with the same data and loss function used in the cold-start stage, confirmed that direct generation performance was negatively impacted by overfitting. Additionally, self-correction ability was limited, yielding only a 0.4 improvement.
>
>     |Methods|Avg Acc|
>     |-|-|
>     |LLaVA-CoT|63.2|
>     |+Self-Correction|63.0|
>     |LLaVA-CoT Cold-Start|62.8|
>     |+Self-Correction|63.2|
> - **Cold-start stage in Sherlock is important.**
>     1. **Cold-start jointly unlocks reasoning and self-correction capabilities.** Our Sherlock-SFT loss (Eq. 3) integrates direct generation and self-correction objectives, transforming a base VLM into a reasoning VLM. Table 2 shows that our cold-start training, using part of LLaVA-CoT dataset annotations, effectively enabled both reasoning (62.2 acc) and self-correction (↑0.8 gain) in the base model.
>     2. **Sherlock-SFT is a better choice.** We posit Sherlock-SFT as a superior SFT paradigm. Fine-tuning Llama3.2V-11B models with identical data, Sherlock-SFT consistently outperformed Vanilla SFT in direct reasoning at both 10k and 20k samples. Additionally, the Sherlock-trained model's performance improved with our internal self-correction prompt at inference, while Vanilla SFT degraded. This demonstrates that introducing a self-correction objective during training significantly enhances direct reasoning, validating our decision to retrain a base model as Sherlock-SFT fully leverages the data for optimal performance.
>
>         |Methods|#Data|Avg Acc|
>         |-|-|-|
>         |Vanilla SFT|10k|61.3|
>         |+Self-Correction|10k|61.2|
>         |Sherlock-SFT|10k|62.2|
>         |+Self-Correction|10k|63.0|
>         |Vanilla SFT|20k|62.0|
>         |+Self-Correction|20k|61.8|
>         |Sherlock-SFT|20k|62.8|
>         |+Self-Correction|20k|63.4|
>
>     The cold-start stage is a **deliberate design choice**, establishing Sherlock as a new paradigm for training reasoning VLMs, rather than a limitation relying on pretrained CoT models.
> ***
> > W2: Unvalidated Noise Injection
> 1. **Quality-gap of our preference data.** The most severe case involves truncating to step 1 with maximum noise 1, leading to a trajectory based on a pure noise image. Conversely, the minimal quality gap occurs with a truncation step of 4 and near-zero noise. In this case, preference responses will exhibit minor differences due to subtle variations in sampling parameters. Since both truncation and noise levels are uniformly random, our preference dataset exhibits a uniform distribution of quality gaps. We'll include examples of generated responses under different noise levels in the revision.
> 2. **Large quality-gap not influence learning signal in Sherlock.** Our self-correction loss (Eq.6) explicitly trains the model to self-correct. Even when the quality gap between response pairs is very large, the objective still provides valuable learning signals, as **it teaches the model how to correct extremely poor outputs into correct ones**. This contrasts with DPO, which maximizes the likelihood of the preferred response while minimizing that of the rejected one, and can be sensitive to overly large quality gaps. As mentioned above, our data construction spans a diverse range of quality gaps, enabling the model to generalize effectively across varying levels of self-correction difficulty.
> ***
> > W3: Limited Comparison with Relevant Work
> 1. **Comprehensive Comparison with Recent SOTA Methods.** We have compared Sherlock against recent SOTA reasoning models: LLaVA-CoT (ICCV 2025), Mulberry (arXiv 2024.12), and LlamaV-o1 (ACL 2025). These baselines, built on the stronger Llama3.2V, are more recent than those suggested by reviewers and use different base models, thus direct comparisons with suggested methods aren't applicable. Suggested papers will be included in the related work.
>     Our experiments sufficiently demonstrate Sherlock's advantages: it consistently outperforms strong baselines in direct reasoning accuracy across eight benchmarks with less than 20% of the annotated data. Furthermore, Sherlock uniquely offers inference-time self-correction with minimal latency and no additional models.
> 2. **Fundamental Differences in Self-Improvement Methodology.** The training supervision and scaling strategy are different between Sherlock and listed paper. The paper's self-training relies on **ground-truth labels** for preference datasets, whereas Sherlock's online stage constructs them **without external supervision or reward signals**. Furthermore, Sherlock's efficient self-correction enables inference-time scaling, a more computationally efficient and effective approach than other methods like majority voting, as shown in Table 3.
> 3. **Performance comparison against LLaVA-Reasoner.** To further validate Sherlock's effectiveness, we reproduced LLaVA-Reasoner (the third listed paper) using its original pipeline but with the stronger Llama3.2V as the base VLM. Sherlock significantly outperformed this LLaVA-Reasoner-DPO across all benchmarks. Notably, CoT prompt to LLaVA-Reasoner showed no improvement on the stronger base VLM, suggesting that older, unstructured CoT methods may not enhance newer VLMs.
>
>     |Model|MMB|MMVet|Hallus|MMMU|MMStar|AI2D|MathV|MME|Avg|
>     |-|-|-|-|-|-|-|-|-|-|
>     |LLaVA-Reasoner-DPO|73.5|56.2|45.9|42.4|57.1|87.6|50.8|2001|60.6|
>     |+CoT|75.5|53.9|43.1|43.9|54.6|87.8|46.0|1875|59.0|
>     |Sherlock|74.6|62.4|48.7|49.7|57.7|89.6|52.0|2197|64.1|
>     |+Self-Correction|77.2|62.6|51.2|50.1|59.0|90.6|54.0|2204|65.4|
> ***
> > W4: Modest Performance Gain over Baselines
> 1. **Performance Gain is Significant and Data-Efficient.** Sherlock's performance gain is significant, especially given its data efficiency. Its average score of 64.1 surpasses recent models like LLaVA-CoT (63.2), Mulberry (60.2), and LlamaV-o1 (63.4). Crucially, Sherlock achieves this using only 20k annotated and 10k unlabeled samples, whereas these competing models were trained on 100k to 260k annotated samples.
>     For a direct comparison, we retrained LLaVA-CoT on the same 20k annotated samples used by Sherlock. The results confirm Sherlock's superior learning efficiency, as it further outperforms LLaVA-CoT. Furthermore, Sherlock's training uniquely enables self-correction. When a self-correction prompt was applied to all models, only Sherlock received a "free lunch" performance boost (64.1 to 65.4). The baselines failed to improve, proving Sherlock successfully activates this capability.
>
>     |Methods|#Data|Avg Acc|
>     |-|-|-|
>     |LLaVA-CoT|100k|63.2|
>     |+Self-Correction|100k|63.0|
>     |LlamaV-o1|175k|63.4|
>     |+Self-Correction|175k|48.2|
>     |LLaVA-CoT-20k|20k|62.0|
>     |+Self-Correction|20k|61.8|
>     |Sherlock|20k|64.1|
>     |+Self-Correction|20k|65.4|
>
> 2. **Sherlock's Training is More Efficient Than Baselines.** Sherlock's training is also considerably less complex than baseline models. Our pipeline uses only SFT and non-RL preference training, whereas methods like LlamaV-o1 employ more computationally intensive stages such as multi-turn conversation training. The table below reports the total A100 GPU hours for each model's entire training process (conducted on 8xA100 GPUs with DeepSpeed Zero-3). The results clearly show Sherlock's training requires substantially less time, highlighting its efficiency.
>
>     |Methods|GPU Hours|
>     |-|-|
>     |LLaVA-CoT|160|
>     |LlamaV-o1|288|
>     |Sherlock|128|
>
> > Minor Weakness
>
> Thanks for the constructive suggestions. We will add the corresponding discussion content to further enhance the completeness of our manuscript.
> ***
> > Q2
>
> Since LLaVA-CoT already generates CoT responses and our prior analysis shows SFT is ineffective on it, we skipped that stage. Instead, we applied Sherlock's offline preference training directly to LLaVA-CoT using the same preference data. The results are presented below.
> Our results show that offline preference training successfully unlocked self-correction in LLaVA-CoT and slightly improved its direct generation. However, comparing this model to Sherlock Offline reveals that cold-starting from a base model is more effective for developing this capability. This may be because LLaVA-CoT's prior exposure to the training data limited the benefit of additional preference training.
>
> |Methods|Avg Acc|
> |-|-|
> |LLaVA-CoT|63.2|
> |+Self-Correction|63.0|
> |LLaVA-CoT Offline|63.6|
> |+Self-Correction|64.2|
> |Sherlock Offline|63.2|
> |+Self-Correction|64.4|
> |Sherlock Iter2|64.1|
> |+Self-Correction|65.4|

---

> > ### Comment · Reviewer_Kt1S · 2025-08-05
> >
> > Thanks for your(s) response. Could the author(s) clarify how the training dataset sizes (e.g., 10K, 20K) were determined? Was any scaling analysis performed (e.g., testing on 10K, 20K, 30K, 40K, etc.) to evaluate the impact of dataset size on model performance? If so, please discuss the trends observed. If not, please justify the chosen sizes.

---

> > > ### Author Response · Authors · 2025-08-05
> > >
> > > Thanks for the valuable question!
> > >
> > > - **If you are referring to the dataset size selection in our paper:** Since annotated data is expensive for training reasoning models, Sherlock is designed to elicit reasoning and self-correction capabilities in VLMs with **minimal annotation**. We use 20k annotated examples in total (10k cold-start + 10k offline). This size was selected based on empirical performance. Specifically, we systematically varied the data set (5k, 10k, 20k) and found that **10k examples in each stage are sufficient** to induce both the CoT template and reliable self-correction behavior.
> > >
> > >    **With only 20k data, Sherlock already outperforms all baselines.** While we expect additional data to yield further improvements, time constraints prevent us from conducting exhaustive scaling experiments. These results will be included in the revised version.
> > >
> > > - **If you are referring to the dataset sizes reported in the rebuttal W1:** For fair comparison with vanilla SFT, we adopt identical splits:
> > >    - **10k split:** the same 10k examples used in our cold-start stage.
> > >    - **20k split:** the union of the 10k cold-start and 10k offline samples, totaling 20k annotations.
> > >
> > >    We therefore include Sherlock-Offline results in the table below and summarize the findings:
> > >
> > >    1. **Scaling helps:** both Vanilla-SFT and Sherlock-SFT improve their direct-generation accuracy as data increases.
> > >    2. **Self-correction emerges only in Sherlock-SFT**, regardless of dataset size; Vanilla-SFT alway fails to learn self-correct bahavior.
> > >    3. **Sherlock-SFT 20k vs. Sherlock-Offline**: with the same annotation budget, offline preference training injects richer learning signals and finer-grained objectives, yielding superior performance on both direct generation and self-correction. Therefore, in Sherlock, we only use 10k data to cold-start the base model, followed by an offline training phase.
> > >
> > >     |Methods|#Data|Avg Acc|
> > >     |-|-|-|
> > >     |Vanilla SFT|10k|61.3|
> > >     |+Self-Correction|10k|61.2$^{\\downarrow 0.1}$|
> > >     |Sherlock-SFT|10k|62.2|
> > >     |+Self-Correction|10k|63.0$^{\\uparrow 0.8}$|
> > >     |Vanilla SFT|20k|62.0|
> > >     |+Self-Correction|20k|61.8$^{\\downarrow 0.2}$|
> > >     |Sherlock-SFT|20k|62.8|
> > >     |+Self-Correction|20k|63.4$^{\\uparrow 0.6}$|
> > >     |Sherlock-Offline|20k|63.2|
> > >     |+Self-Correction|20k|64.4$^{\\uparrow 1.2}$|

---

> > > > ### Comment · Reviewer_Kt1S · 2025-08-05
> > > >
> > > > Yes, it refers to the dataset size selection in your(s) paper. Thanks for your(s) prompt response. I have no more questions about this work. will raise my score.

---

> > > > > ### Author Response · Authors · 2025-08-05
> > > > >
> > > > > Thank you for your response and for increasing your rating. We truly appreciate your valuable insights, which have helped us improve the quality of our work. We will incorporate the suggested experiments and discussion in the revised version. Thank you again for your support of our paper!

---

### Decision · Program_Chairs · 2025-09-17

**Decision:**

Accept (poster)

**Comment:**

This paper presents a self-correction and iterative self-improvement framework for LVLMs to improve the reasoning ability under the data efficiency setting. The authors propose a three-stage training framework, including SFT, offline DPO, and iterative online preference training. All reviewers agree that this work's contribution is strong with the following contributions.
1. Citing Reviewer aRRH (score 4): The paper addresses a timely and important problem—enhancing reasoning and self-correction capabilities in VLMs.
2. Citing Reviewer 72iE (score 4):  Sherlock requires only 20k randomly annotated data for initialization, then improves unsupervised via online iterative self-generated preference pairs.

The majority of the reviewers' concerns were the missed implementation details or limited ablation studies. Below are summarized key points during the reviewer-author discussion:
1. Reviewers have concerns about the main experimental results being conducted on small-sized models. The authors provide the additional results of VL-Rethinker-72B to address this concern.
2. Reviewers also have concerns about the comparison with more recent and relevant work. The authors discuss the results of  LLaVA-CoT and LLaVAV-o1.
3. The authors provide more technical details in rebuttal, including the necessity of the SFT stage, the generation process,  model choices, and the discussion with the dynamic parameter $\beta$.

After the author-reviewer discussion period, all reviewers gave positive ratings, and I recommend acceptance. The authors are encouraged to add the additional experimental results and implementation details in their final version.